# Contextual Recommendations and Low-Regret Cutting-Plane Algorithms

Sreenivas Gollapudi,  Guru Guruganesh, Kostas Kollias, Pasin Manurangsi, Renato Paes Leme, and Jon Schneider

Google Research

## Abstract

We consider the following variant of contextual linear bandits motivated by routing applications in navigational engines and recommendation systems. We wish to learn a hidden $d$-dimensional value $w^*$. Every round, we are presented with a subset $\mathcal{X}_t \subseteq \mathbb{R}^d$ of possible actions. If we choose (i.e. recommend to the user) action $x_t$, we obtain utility $\langle x_t, w^* \rangle$ but only learn the identity of the best action $\arg\max_{x \in \mathcal{X}_t} \langle x, w^* \rangle$.

We design algorithms for this problem which achieve regret $O(d \log T)$ and $\exp(O(d \log d))$. To accomplish this, we design novel cutting-plane algorithms with low "regret" – the total distance between the true point $w^*$ and the hyperplanes the separation oracle returns.

We also consider the variant where we are allowed to provide a list of several recommendations. In this variant, we give an algorithm with $O(d^2 \log d)$ regret and list size $\mathrm{poly}(d)$. Finally, we construct nearly tight algorithms for a weaker variant of this problem where the learner only learns the identity of an action that is better than the recommendation. Our results rely on new algorithmic techniques in convex geometry (including a variant of Steiner's formula for the centroid of a convex set) which may be of independent interest.

## 1   Introduction

Consider the following problem faced by a geographical query service (e.g. Google Maps). When a user searches for a path between two endpoints, the service must return one route out of a set of possible routes. Each route has a multidimensional set of features associated with it, such as (i) travel time, (ii) amount of traffic, (iii) how many turns it has, (iv) total distance, etc. The service must recommend one route to the user, but doesn't a priori know how the user values these features relative to one another. However, when the service recommends a route, the service can observe some feedback from the user: whether or not the user followed the recommended route (and if not, which route the user ended up taking). How can the service use this feedback to learn the user's preferences over time?

Similar problems are faced by recommendation systems in general, where every round a user arrives accompanied by some contextual information (e.g. their current search query, recent activity, etc.), the system makes a recommendation to the user, and the system can observe the eventual action (e.g. the purchase of a specific item) by the user. These problems can be viewed as specific cases of a variant of linear contextual bandits that we term *contextual recommendation*.

In contextual recommendation, there is a hidden vector $w^* \in \mathbb{R}^d$ (e.g. representing the values of the user for different features) that is unknown to the learner. Every round $t$ (for $T$ rounds), the learner is presented with an adversarially chosen (and potentially very large) set of possible actions $\mathcal{X}_t$. Each

35th Conference on Neural Information Processing Systems (NeurIPS 2021).

element $x_t$ of $\mathcal{X}_t$ is also an element of $\mathbb{R}^d$ (visible to the learner); playing action $x_t$ results in the learner receiving a reward of $\langle x_t, w^* \rangle$. The learner wishes to incur low regret compared to the best possible strategy in hindsight – i.e. the learner wishes to minimize

$$\text{Reg} = \sum_{t=1}^{T} \left( \langle x_t^*, w^* \rangle - \langle x_t, w^* \rangle \right), \tag{1}$$

where $x_t^* = \arg\max_{x \in \mathcal{X}_t} \langle x, w^* \rangle$ is the best possible action at time $t$. In our geographical query example, this regret corresponds to the difference between the utility of a user that always blindly follows our recommendation and the utility of a user that always chooses the optimal route.

Thus far this agrees with the usual set-up for contextual linear bandits (see e.g. [4]). Where contextual recommendation differs from this is in the feedback available to the learner: whereas classically in contextual linear bandits the learner learns (a possibly noisy version of) the reward they receive each round, in contextual recommendation the learner instead learns *the identity of the best arm $x_t^*$*. This altered feedback makes it difficult to apply existing algorithms for linear contextual bandits. In particular, algorithms like LINUCB and LIN-Rel [1, 4] all require estimates of $\langle x_t, w^* \rangle$ in order to learn $w^*$ over time, and our feedback prevents us from obtaining any such absolute estimates.

In this paper we design low-regret algorithms for this problem. We present two algorithms for this problem: one with regret $O(d \log T)$ and one with regret $\exp(O(d \log d))$ (Theorems 4.1 and 4.2). Note that both regret guarantees are independent of the number of offered actions $|\mathcal{X}_t|$ (the latter even being independent of the time horizon $T$). Moreover both of these algorithms are efficiently implementable given an efficient procedure for optimizing a linear function over the sets $\mathcal{X}_t$. This condition holds e.g. in the example of recommending shortest paths that we discussed earlier.

In addition to this, we consider two natural extensions of contextual recommendation where the learner is allowed to recommend a bounded subset of actions instead of just a single action (as is often the case in practice). In the first variant, which we call *list contextual recommendation*, each round the learner recommends a set of at most $L$ (for some fixed $L$) actions to the learner. The learner still observes the user's best action each round, but the loss of the learner is now the difference between the utility of the best action for the user and the best action offered by the learner (capturing the difference in utility between a user playing an optimal action and a user that always chooses the best action the learner offers).

In list contextual recommendation, the learner has the power to cover multiple different user preferences simultaneously (e.g. presenting the user with the best route for various different measures). We show how to use this power to construct an algorithm for the learner which offers $\text{poly}(d)$ actions each round and obtain a total regret of $\text{poly}(d)$.

In the second variant, we relax an assumption of both previous models: that the user will always choose their best possible action (and hence that we will observe their best possible action). To relax this assumption, we also consider the following weaker version of contextual recommendation we call *local contextual recommendation*.

In this problem, the learner again recommends a set of at most $L$ actions to the learner (for some $L > 1$)[1]. The user then chooses an action which is at least as good as the best action in our list, and we observe this action. In other words, we assume the learner at least looks at all the options we offer, so if they choose an external option, it must be better than any offered option (but not necessarily the global optimum). Our regret in this case is the difference between the total utility of a learner that always follows the best recommendation in our list and the total utility of a learner that always plays their optimal action[2].

Let $A = \max_t |\mathcal{X}_t|$ be a bound on the total number of actions offered in any round, and let $\gamma = A/(L-1)$. We construct algorithms for local contextual recommendation with regret $O(\gamma d \log T)$ and $\gamma \cdot \exp(O(d \log d))$. We further show that the first bound is "nearly tight" (up to $\text{poly}(d)$ factors) in some regimes; in particular, we demonstrate an instance where $L = 2$ and $K = 2^{\Omega(d)}$

---

[1]Unlike in the previous two variants, it is important in local contextual recommendation that $L > 1$; if $L = 1$ then the user can simply report the action the learner recommended and the learner receives no meaningful feedback.

[2]In fact, our algorithms all work for a slightly stronger notion of regret, where the benchmark is the utility of a learner that always follows the *first* (i.e. a specifically chosen) recommendation on our list. With this notion of regret, contextual recommendation reduces to local contextual recommendation with $L = \max |\mathcal{X}_t|$.

where any algorithm must incur regret at least $\min(2^{\Omega(d)}, \Omega(T))$. The results for local contextual recommendation are included in the Supplementary Material.

## 1.1 Low-regret cutting plane methods and contextual search

To design these low-regret algorithms, we reduce the problem of contextual recommendation to a geometric online learning problem (potentially of independent interest). We present two different (but equivalent) viewpoints on this problem: one motivated by designing separation-oracle-based algorithms for convex optimization, and the other by contextual search.

### 1.1.1 Separation oracles and cutting-plane methods

Separation oracle methods (or "cutting-plane methods") are an incredibly well-studied class of algorithms for linear and convex optimization. For our purposes, it will be convenient to describe cutting-plane methods as follows.

Let $\mathsf{B} = \{w \in \mathbb{R}^d \mid \|w\| \leq 1\}$ be the unit ball in $\mathbb{R}^d$. We are searching for a hidden point $w^* \in \mathsf{B}$. Every round we can choose a point $p_t \in \mathsf{B}$ and submit this point to a *separation oracle*. The separation oracle then returns a half-space separating $p_t$ from $w^*$; in particular, the oracle returns a direction $v_t$ such that $\langle w^*, v_t \rangle \geq \langle p_t, v_t \rangle$.

Traditionally, cutting-plane algorithms have been developed to minimize the number of calls to the separation oracle until the oracle returns a hyperplane that passes within some distance $\delta$ of $w^*$. For example, the ellipsoid method (which always queries the center of the currently-maintained ellipse) has the guarantee that it makes at most $O(d^2 \log 1/\delta)$ oracle queries before finding such a hyperplane.

In our setting, instead of trying to minimize the number of separation oracle queries before finding a "close" hyperplane, we would like to minimize the total (over all $T$ rounds) distance between the returned hyperplanes and the hidden point $w^*$. That is, we would like to minimize the expression

$$\mathrm{Reg}' = \sum_{t=1}^{T} \left( \langle w^*, v_t \rangle - \langle p_t, v_t \rangle \right). \tag{2}$$

Due to the similarity between (2) and (1), we call this quantity the *regret* of a cutting-plane algorithm. We show that, given any low-regret cutting-plane algorithm, there exists a low-regret algorithm for contextual recommendation.

**Theorem 1.1** (Restatement of Theorem 3.1). *Given a low-regret cutting-plane algorithm $\mathcal{A}$ with regret $\rho$, we can construct an $O(\rho)$-regret algorithm for contextual recommendation.*

This poses a natural question: what regret bounds are possible for cutting-plane methods? One might expect guarantees on existing cutting-plane algorithms to transfer over to regret bounds, but interestingly, this does not appear to be the case. In particular, most existing cutting-plane methods and analysis suffers from the following drawback: even if the method is likely to find a hyperplane within distance $\delta$ relatively quickly, there is no guarantee that subsequent calls to the oracle will return low-regret hyperplanes.

In this paper, we will show how to design low-regret cutting-plane methods. Although our final algorithms will bear some resemblance to existing cutting-plane algorithms (e.g. some involve cutting through the center-of-gravity of some convex set), our analysis will instead build off more recent work on the problem of *contextual search*.

### 1.1.2 Contextual search

Contextual search is an online learning problem initially motivated by applications in pricing [6]. The basic form of contextual search can be described as follows. As with the previously mentioned problems, there is a hidden vector $w^* \in [0, 1]^d$ that we wish to learn over time. Every round the adversary provides the learner with a vector $v_t$ (the "context"). In response, the learner must guess the value of $\langle v_t, w^* \rangle$, submitting a guess $y_t$. The learner then incurs a loss of $|\langle v_t, w^* \rangle - y_t|$ (the distance between their guess and the true value of the inner product), but only learns whether $\langle v_t, w^* \rangle$ is larger or smaller than their guess.

The problem of designing low-regret cutting plane methods can be interpreted as a "context-free" variant of contextual search. In this variant, the learner is no longer provided the context $v_t$ at the beginning of each round, and instead of guessing the value of $\langle v_t, w^* \rangle$, they are told to directly

submit a guess $p_t$ for the point $w^*$. The context $v_t$ is then revealed to them *after* they submit their guess, where they are then told whether $\langle p_t, w^* \rangle$ is larger or smaller than $\langle v_t, w^* \rangle$ and incur loss $|\langle v_t, w^* \rangle - \langle p_t, w^* \rangle|$. Note that this directly corresponds to querying a separation oracle with the point $p_t$, and the separation oracle returning either the halfspace $v_t$ (in the case that $\langle w^*, v_t \rangle \geq \langle w^*, p_t \rangle$) or the halfspace $-v_t$ (in the case that $\langle w^*, v_t \rangle \leq \langle w^*, p_t \rangle$).

One advantage of this formulation is that (unlike in standard analyses of cutting-plane methods) the total loss in contextual search directly matches the expression in (2) for the regret of a cutting-plane method. In fact, were there to already exist an algorithm for contextual search which operated in the above manner – guessing $\langle v_t, w^* \rangle$ by first approximating $w^*$ and then computing the inner product – we could just apply this algorithm verbatim and get a cutting-plane method with the same regret bound. Unfortunately, both the algorithms of [7] and [6] explicitly require knowledge of the direction $v_t$.

This formulation also raises an interesting subtlety in the power of the separation oracle: specifically, whether the direction $v_t$ is fixed (up to sign) ahead of time or is allowed to depend on the point $p$. Specifically, we consider two different classes of separation oracles. For *(strong) separation oracles*, the direction $v_t$ is allowed to freely depend on the point $p_t$ (as long as it is indeed true that $\langle w^*, v_t \rangle \geq \langle p_t, v_t \rangle$). For *weak separation oracles*, the adversary fixes a direction $u_t$ at the beginning of the round, and then returns either $v_t = u_t$ or $v_t = -u_t$ (depending on the sign of $\langle w^* - p_t, u_t \rangle$). The strong variant is most natural when comparing to standard separation oracle guarantees (and is necessary for the reduction in Theorem 1.1), but for many standalone applications (especially those motivated by contextual search) the weak variant suffices. In addition, the same techniques we use to construct a cutting-plane algorithm for weak separation oracles will let us design low-regret algorithms for list contextual recommendation.

## 1.2   Our results and techniques

We design the following low-regret cutting-plane algorithms:

1. An $\exp(O(d \log d))$-regret cutting-plane algorithm for strong separation oracles.

2. An $O(d \log T)$-regret cutting-plane algorithm for strong separation oracles.

3. An $O(\mathrm{poly}(d))$-regret cutting-plane algorithm for weak separation oracles.

All three algorithms are efficiently implementable (in $\mathrm{poly}(d, T)$ time). Through Theorem 1.1, points (1) and (2) immediately imply the algorithms with regret $\exp(O(d))$ and $O(d \log T)$ for contextual recommendation. Although we do not have a blackbox reduction from weak separation oracles to algorithms for list contextual recommendation, we show how to apply the same ideas in the algorithm in point (3) to construct an $O(d^2 \log d)$-regret algorithm for list contextual recommendation with $L = \mathrm{poly}(d)$.

To understand how these algorithms work, it is useful to have a high-level understanding of the algorithm of [7] for contextual search. That algorithm relies on a multiscale potential function the authors call the *Steiner potential*. The Steiner potential at scale $r$ is given by the expression $\mathsf{Vol}(K_t + r\mathsf{B})$, where $K_t$ (the "knowledge set") is the current set of possibilities for the hidden point $w^*$, $\mathsf{B}$ is the unit ball, and addition denotes Minkowsi sum; in other words, this is the volume of the set of points within distance $r$ of $K_t$. The authors show that by choosing their guess $y_t$ carefully, they can decrease the $r$-scale Steiner potential (for some $r$ roughly proportional to the width of $K_t$ in the current direction $v_t$) by a constant factor. In particular, they show that this is achieved by choosing $y_t$ so to divide the expanded set $K_t + r\mathsf{B}$ exactly in half by volume. Since the Steiner potential at scale $r$ is bounded below by $\mathsf{Vol}(r\mathsf{B})$, this allows the authors to bound the total number of mistakes at this scale. (A more detailed description of this algorithm is provided in Section 2.2).

In the separation oracle setting, we do not know $v_t$ ahead of time, and thus cannot implement this algorithm as written. For example, we cannot guarantee our hyperplane splits $K_t + r\mathsf{B}$ exactly in half. We partially work around this by using (approximate variants of) Grunbaum's theorem, which guarantees that any hyperplane through the center-of-gravity of a convex set splits that convex set into two pieces of roughly comparable volume. In other words, everywhere where the contextual search algorithm divides the volume of $K_t + r\mathsf{B}$ in half, Grunbaum's theorem implies we obtain comparable results by choosing any hyperplane passing through the center-of-gravity of $K_t + r\mathsf{B}$.

Unfortunately, we still cannot quite implement this in the separation oracle setting, since the choice of $r$ in the contextual search algorithm depends on the input vector $v_t$. Nonetheless, by modifying the analysis of contextual search we can still get some guarantees via simple methods of this form. In particular we show that always querying the center-of-gravity of $K_t$ (alternatively, the center of the John ellipsoid of $K_t$) results in an $\exp(O(d \log d))$-regret cutting-plane algorithm, and that always querying the center of gravity of $K_t + \frac{1}{T}\mathsf{B}$ results in an $O(d \log T)$-regret cutting-plane algorithm.

Our cutting-plane algorithm for weak separation oracles requires a more nuanced understanding of the family of sets of the form $K_t + r\mathsf{B}$. This family of sets has a number of surprising algebraic properties. One such property (famous in convex geometry and used extensively in earlier algorithms for contextual search) is *Steiner's formula*, which states that for any convex $K$, $\mathsf{Vol}(K + r\mathsf{B})$ is actually a polynomial in $r$ with nonnegative coefficients. These coefficients are called *intrinsic volumes* and capture various geometric measures of the set $K$ (including the volume and surface area of $K$).

There exists a lesser-known analogue of Steiner's formula for the center-of-gravity of $K + r\mathsf{B}$, which states that each coordinate of $\mathsf{cg}(K + r\mathsf{B})$ is a rational function of degree at most $d$; in other words, the curve $\mathsf{cg}(K + r\mathsf{B})$ for $r \in [0, \infty)$ is a rational curve. Moreover, this variant of Steiner's formula states that each point $\mathsf{cg}(K + r\mathsf{B})$ can be written as a convex combination of $d + 1$ points contained within $K$ known as the *curvature centroids* of $K$. Motivated by this, we call the curve $\rho_K(r) = \mathsf{cg}(K + r\mathsf{B})$ the *curvature path* of $K$.

Since the curvature path $\rho_K$ is both bounded in algebraic degree and bounded in space (having to lie within the convex hull of the curvature centers), we can bound the total length of the curvature path $\rho_K$ by a polynomial in $d$ (since it is bounded in degree, each component function of $\rho_K$ can switch from increasing to decreasing a bounded number of times). This means that we can discretize the curvature path to within precision $\varepsilon$ while only using $\mathrm{poly}(d)/\varepsilon$ points on the path.

Our algorithms against weak separation oracles and for list contextual recommendation both make extensive use of such a discretization. For example, we show that in order to construct a low-regret algorithm against a weak separation oracle, it suffices to discretize $\rho_{K_t}$ into $O(d^4)$ points and then query a random point; with probability at least $O(d^{-4})$, we will closely enough approximate the point $\rho(r) = \mathsf{cg}(K + r\mathsf{B})$ that our above analogue of contextual search would have queried. We show this results in $\mathrm{poly}(d)$ total regret[3]. A similar strategy works for list contextual recommendation: there we discretize the curvature path for the knowledge set $K_t$ into $\mathrm{poly}(d)$ candidate values for $w^*$, and then submit as our set of actions the best response for each of these candidates.

**Related work**    We include a survey of related work in the Supplementary Material.

## 2    Model and preliminaries

We begin by briefly reviewing the problems of contextual recommendation and designing low-regret cutting plane algorithms. In all of the below problems, $\mathsf{B} = \{w \in \mathbb{R}^d \mid \|w\|_2 \le 1\}$ is the ball of radius 1 (and generally, all vectors we consider will be bounded to lie in this ball).

**Contextual recommendation.**    In *contextual recommendation* there is a hidden point $w^* \in \mathsf{B}$. Each round $t$ (for $T$ rounds) we are given a set of possible actions $\mathcal{X}_t \subseteq \mathsf{B}$. If we choose action $x_t \in \mathcal{X}_t$ we obtain reward $\langle x_t, w^* \rangle$ (but do not learn this value). Our feedback is $x_t^* = \arg\max_{x \in \mathcal{X}_t} \langle x, w^* \rangle$, the identity of the best action[4]. Our goal is to minimize the total expected regret $\mathbb{E}[\mathrm{Reg}] = \mathbb{E}\left[\sum_{t=1}^{T} \langle x_t^* - x_t, w^* \rangle\right]$. Note that since the feedback is deterministic, this expectation is only over the randomness of the learner's algorithm.

It will be useful to establish some additional notation for discussing algorithms for contextual recommendation. We define the *knowledge set* $K_t$ to be the set of possible values for $w^*$ given the knowledge we have obtained by round $t$. Note that the knowledge set $K_t$ is always convex, since the feedback we receive each round (that $\langle x_t^*, w^* \rangle \ge \langle x_t, w^* \rangle$ for all $x \in \mathcal{X}_t$) can be written as an

---

[3]The reason this type of algorithm does not work against strong separation oracles is that each point in this discretization could return a different direction $v_t$, in turn corresponding to a different value of $r$

[4]If this argmax is multi-valued, the adversary may arbitrarily return any element of this argmax.

intersection of several halfspaces (and the initial knowledge set $K_1 = \mathsf{B}$ is convex). In fact, we can say more.

Given a $w \in K_t$, let $\mathsf{BR}_t(w) = \arg\max_{x \in \mathcal{X}_t} \langle x, w \rangle$ be the set of optimal actions in $\mathcal{X}_t$ if the hidden point was $w$. Then: $K_{t+1} = \{w \in K_t | x_t^* \in \mathsf{BR}_t(w)\}$.

We also consider two other variants of contextual recommendation in this paper (*list contextual recommendation* and *local contextual recommendation*). A formal definition can be found in Sections 5 and the full version in the supplementary materials.

**Designing low-regret cutting-plane algorithms.** In a *low-regret cutting-plane algorithm*, we again have a hidden point $w^* \in \mathsf{B}$. Each round $t$ (for $T$ rounds) we can query a separation oracle with a point $p_t$ in B. The separation oracle then provides us with an adversarially chosen direction $v_t$ (with $\|v_t\| = 1$) that satisfies $\langle w^*, v_t \rangle \geq \langle p_t, v_t \rangle$. The regret in round $t$ is equal to $\langle w^* - p_t, v_t \rangle$, and our goal is to minimize the total expected regret $\mathbb{E}[\mathrm{Reg}] = \mathbb{E}\left[\sum_{t=1}^T \langle w^* - p_t, v_t \rangle\right]$. Again, since the feedback is deterministic, the expectation is only over the randomness of the learner's algorithm.

As with contextual recommendation, it will be useful to consider the knowledge set $K_t$, consisting of possibilities for $w^*$ which are still feasible by the beginning of round $t$. Again as with contextual recommendation, $K_t$ is always convex; here we intersect $K_t$ with the halfspace provided by the separation oracle every round (i.e. $K_{t+1} = K_t \cap \{\langle w - p_t, v_t \rangle \geq 0\}$).

Unless otherwise specified, the separation oracle can arbitrarily choose $v_t$ as a function of the query point $p_t$. For obtaining low-regret algorithms for list contextual recommendation, it will be useful to consider a variant of this problem where the separation oracle must commit to $v_t$ (up to sign) at the beginning of round $t$. Specifically, at the beginning of round $t$ (before observing the query point $p_t$), the oracle fixes a direction $u_t$. Then, on query $p_t$, the separation oracle returns the direction $v_t = u_t$ if $\langle w - p_t, u_t \rangle \geq 0$, and the direction $v_t = -u_t$ otherwise. We call such a separation oracle a *weak separation oracle*; an algorithm that only works against such separation oracles is a *low-regret cutting-plane algorithm for weak separation oracles*. Note that this distinction only matters when the learner is using a randomized algorithm; if the learner is deterministic, the adversary can predict all the directions $v_t$ in advance.

## 2.1 Convex geometry preliminaries and notation

We will denote by $\mathsf{Conv}_d$ the collection of all convex bodies in $\mathbb{R}^d$. Given a convex body $K \in \mathsf{Conv}_d$, we will use $\mathsf{Vol}(K) = \int_K 1 dx$ to denote its volume (the standard Lebesgue measure). Given two sets $K$ and $L$ in $\mathbb{R}^d$, their Minkowski sum is given by $K + L = \{x + y; x \in K, y \in L\}$. Let $\mathsf{B}^d$ denote the unit ball in $\mathbb{R}^d$, let $\mathbb{S}^{d-1} = \{x \in \mathbb{R}^d; \|x\|_2 = 1\}$ denote the unit sphere in $\mathbb{R}^d$ and let $\kappa_d = \mathsf{Vol}(\mathsf{B}^d)$ be the volume of the $i$-th dimensional unit ball. When clear from context, we will omit the superscripts on $\mathsf{B}^d$ and $\mathbb{S}^{d-1}$.

We will write $\mathsf{cg}(K) = (\int_K x dx)/(\int_K 1 dx)$ to denote the *center of gravity* (alternatively, *centroid*) of $K$. Given a direction $u \in \mathbb{S}^{d-1}$ and convex set $K \in \mathsf{Conv}_d$ we define the width of $K$ in the direction $u$ as: $\mathsf{width}(K; u) = \max_{x \in K} \langle u, x \rangle - \min_{x \in K} \langle u, x \rangle$.

**Approximate Grunbaum and John's Theorem** Finally, we state two fundamental theorems in convex geometry. Grunbaum's Theorem bounds the volume of the convex set in each side of a hyperplane passing through the centroid. For our purposes it will be also important to bound a cut that passes near, but not exactly at the centroid. The bound given in the following paragraph comes from a direct combination of Lemma B.4 and Lemma B.5 in Bubeck et al. [3].

We will use the notation $H_u(p) = \{x \mid \langle x, u \rangle = \langle p, u \rangle\}$ to denote the halfspace passing through $p$ with normal vector $u$. Similarly, we let $H_u^+(p) = \{x \mid \langle x, u \rangle \geq \langle p, u \rangle\}$.

**Theorem 2.1** (Approximate Grunbaum [2, 3]). *Let* $K \in \mathsf{Conv}_d$, $c = \mathsf{cg}(K)$ *and* $u \in \mathbb{S}^{d-1}$. *Then consider the semi-space* $H_+ = \{x \in \mathbb{R}^d; \langle u, x - c \rangle \geq t\}$ *for some* $t \in \mathbb{R}_+$. *Then:*

$$\frac{\mathsf{Vol}(K \cap H_+)}{\mathsf{Vol}(K)} \geq \frac{1}{e} - \frac{2t(d+1)}{\mathsf{width}(K;u)}$$

John's theorem shows that for any convex set $K \in \mathsf{Conv}_d$, we can find an ellipsoid $E$ contained in $K$ such that $K$ is contained in (some translate of) a dilation of $E$ by a factor of $d$.

**Theorem 2.2** (John's Theorem). *Given $K \in \mathsf{Conv}_d$, there is a point $q \in K$ and an invertible linear transformation $A : \mathbb{R}^d \to \mathbb{R}^d$ such that $q + \mathsf{B} \subseteq A(K) \subseteq q + d\mathsf{B}$. We call the ellipsoid $E = A^{-1}(q + \mathsf{B})$ the John ellipsoid of $K$.*

## 2.2 Contextual search

In this section, we briefly sketch the algorithm and analysis of [7] for the standard contextual search problem. We will never use this algorithm directly, but many pieces of the analysis will prove useful in our constructions of low-regret cutting-plane algorithms.

Recall that in contextual search, each round the learner is given a direction $v_t$. The learner is trying to learn the location of a hidden point $w^*$, and at time $t$ has narrowed down the possibilities of $w^*$ to a knowledge set $K_t$. The algorithm of [7] runs the following steps: 1. Compute the width $w = \mathsf{width}(K_t; v_t)$ of $K_t$ in the direction $v_t$. 2. Let $r = 2^{\lceil \lg(w/10d) \rceil}$ (rounding $w/10d$ to a nearby power of two). 3. Consider the set $\tilde{K} = K_t + r\mathsf{B}$. Choose $y_t$ so that the hyperplane $H = \{w \mid \langle v_t, w \rangle = y_t\}$ divides the set $\tilde{K}$ into two pieces of equal volume.

We can understand this algorithm as follows. Classic cutting-plane methods try to decrease $\mathsf{Vol}(K_t)$ by a constant factor every round (arguing that this decrease can only happen so often before one of our hyperplanes passes within some small distance to our feasible region). The above algorithm can be thought of as a multi-scale variant of this approach: they show that if we incur loss $w \approx dr$ in a round (since loss in a round is at most the width), the potential function $\mathsf{Vol}(K_t + r\mathsf{B})$ must decrease by a constant factor. Since $\mathsf{Vol}(K_t + r\mathsf{B}) \geq \mathsf{Vol}(r\mathsf{B}) = r^d \kappa_d$, we can incur a loss of this size at most $O(d \log(2/r))$ times. Summing over all possible discretized values of $r$ (i.e. powers of 2 less than 1), we arrive at an $O(d \log d)$ regret bound.

# 3 From Cutting-Plane Algorithms to Contextual Recommendation

We begin by proving a reduction from designing low-regret cutting plane algorithms to contextual recommendation. Specifically, we will show that given a regret $\rho$ cutting-plane algorithm, we can use it to construct an $O(\rho)$-regret algorithm for contextual recommendation.

Note that while these two problems are similar in many ways (e.g. they both involve searching for an unknown point $w^*$), they are not completely identical. Among other things, the formulation of regret although similar is qualitatively different between the two problems (i.e. between expressions (1) and (2)). In particular, in contextual recommendation, the regret each round is $\langle x_t^* - x_t, w^* \rangle$, whereas for cutting-plane algorithms, the regret is given by $\langle w^* - p_t, v_t \rangle$. Nonetheless, we will be able to relate these two notions of regret by considering a separation oracle that always returns a halfspace in the direction of $x_t^* - x_t$.

**Theorem 3.1.** *Given a low-regret cutting-plane algorithm $\mathcal{A}$ with regret $\rho$, we can construct an $O(\rho)$-regret algorithm for contextual recommendation.*

The reduction in Theorem 3.1 is efficient as long as we have an efficient method for optimizing a linear function over $\mathcal{X}_t$ (i.e. for computing $\mathsf{BR}_t(w)$). In particular, this means that this reduction can be practical even in settings where $\mathcal{X}_t$ may be combinatorially large (e.g. the set of $s$-$t$ paths in some graph).

Note also that this reduction *does not* work if $\mathcal{A}$ is only low-regret against weak separation oracles. This is since the direction $v_t$ we choose does depend non-trivially on the point $p_t$ (in particular, in the reduction we choose $x_t \in \mathsf{BR}_t(p_t)$). Later in Section 5.3, we will see how to use ideas from designing cutting-plane methods for weak separation oracles to construct low-regret algorithms for *list* contextual recommendation – however we do not have a black-box reduction in that case, and our construction will be more involved.

# 4 Designing Low-Regret Cutting-Plane Algorithms

In this section we will describe how to construct low-regret cutting-plane algorithms for strong separation oracles.

### 4.1 An $\exp(O(d \log d))$-regret cutting-plane algorithm

We begin by showing that always querying the center of the John ellipsoid of $K_t$ leads to a $\exp(O(d \log d))$-regret cutting-plane algorithm. Interestingly, although this corresponds to the classical ellipsoid algorithm, our analysis proceeds along the lines of the analysis of the contextual search algorithm summarized in Section 2.2. In particular, we show that for each round $t$, there's some $r$ (roughly proportional to the current width) where $\mathsf{Vol}(K_t + rB)$ decreases by a multiplicative factor of $(1 - d^{-O(d)})$. Doing so allows us to prove the following theorem.

**Theorem 4.1.** *The cutting-plane algorithm which always queries the center of the John ellipsoid of $K_t$ incurs $\exp(O(d \log d))$ regret.*

The remaining algorithms we study will generally query the center-of-gravity of some convex set, as opposed to the center of the John ellipsoid. This leads to the following natural question: what is the regret of the cutting-plane algorithm which always queries the center-of-gravity of $K_t$?

Kannan, Lovasz, and Simonovits (Theorem 4.1 of [5]) show that it is possible to choose an ellipsoid $E$ satisfying $E \subseteq K \subseteq dE$ such that $E$ is centered at $\mathsf{cg}(K)$, so our proof of Theorem 4.1 shows that this algorithm is also an $\exp(O(d \log d))$ algorithm. However, for both this algorithm and the ellipsoid algorithm of Theorem 4.1, we have no non-trivial lower bound on the regret. It is an interesting open question to understand what regret these algorithms actually obtain (for example, do either of these algorithms achieve $\mathrm{poly}(d)$ regret?).

### 4.2 An $O(d \log T)$-regret cutting-plane algorithm

We will now show how to obtain an $O(d \log T)$-regret cutting plane algorithm. Our algorithm will simply query the center-of-gravity of $K_t + \frac{1}{T}B$ each round. The advantage of doing this is that we only need to examine one scale of the contextual search potential (namely the value of $\mathsf{Vol}(K_t + \frac{1}{T}B)$). By doing this, we prove that the above algorithm achieves $O(d \log T)$ regret.

**Theorem 4.2.** *The cutting-plane algorithm which queries the point $p_t = \mathsf{cg}\left(K_t + \frac{1}{T}B\right)$ incurs $O(d \log T)$ regret.*

## 5 List contextual recommendation, weak separation oracles, and the curvature path

In this section, we present two algorithms: 1. a $\mathrm{poly}(d)$ expected regret cutting-plane algorithm for weak separation oracles, and 2. an $O(d^2 \log d)$ regret algorithm for list contextual recommendation with list size $L = \mathrm{poly}(d)$.

The unifying feature of both algorithms is that they both involve analyzing a geometric object we call the *curvature path* of a convex body. The *curvature path* of $K$ is a bounded-degree rational curve contained within $K$ that connects the center-of-gravity $\mathsf{cg}(K)$ with the Steiner point ($\lim_{r \to \infty} \mathsf{cg}(K + rB)$) of $K$.

In Section 5.1 we formally define the curvature path and demonstrate how to bound its length. In Section 5.2, we show that randomly querying a point on a discretization of the curvature path leads to a $\mathrm{poly}(d)$ regret cutting-plane algorithm for weak separation oracles. Finally, in Section 5.1, we show how to transform a discretization of the curvature path of the knowledge set into a list of actions for list contextual recommendation, obtaining a low regret algorithm.

### 5.1 The curvature path

An important fact (driving some of the recent results in contextual search, e.g. [6]) is the fact that the volume $\mathsf{Vol}(K + rB)$ is a $d$-dimensional polynomial in $r$. This fact is known as the Steiner formula:

$$\mathsf{Vol}(K + rB) = \sum_{i=0}^{d} V_{d-i}(K) \kappa_i r^i \tag{3}$$

After normalization by the volume of the unit ball, the coefficients of this polynomial correspond to the *intrinsic volumes* of $K$. The intrinsic volumes are a family of $d+1$ functionals $V_i : \mathsf{Conv}_d \to \mathbb{R}_+$ for $i = 0, 1, \ldots, d$ that associate for each convex $K \in \mathsf{Conv}_d$ a non-negative value. Some of these

functionals have natural interpretations: $V_d(K)$ is the standard volume $\mathsf{Vol}(K)$, $V_{d-1}(K)$ is the surface area, $V_1(K)$ is the average width and $V_0(K)$ is 1 whenever $K$ is non-empty and 0 otherwise.

There is an analogue of the Steiner formula for the centroid of $K + r\mathsf{B}$, showing that it admits a description as a vector-valued rational function. More precisely, there exist $d + 1$ functions $c_i : \mathsf{Conv}_d \to \mathbb{R}^d$ for $0 \le i \le d$ such that:

$$\mathsf{cg}(K + r\mathsf{B}) = \frac{\sum_{i=0}^d V_{d-i}(K)\kappa_i r^i \cdot c_i(K)}{\sum_{i=0}^d V_{d-i}(K)\kappa_i r^i} \tag{4}$$

The point $c_0(K) \in K$ corresponds to the usual centroid $\mathsf{cg}(K)$ and $c_d(K)$ corresponds to the Steiner point. The functionals $c_i$ are called *curvature centroids* since they can be computing by integrating a certain curvature measures associated with a convex body (a la Gauss-Bonnet). We refer to Section 5.4 in Schneider [8] for a more thorough discussion discussion. For our purposes, however, the only important fact will be that each curvature centroid $c_i(K)$ is guaranteed to lie within $K$ (note that this is not at all obvious from their definition).

Motivated by this, given a convex body $K \subseteq \mathbb{R}^d$ we define its *curvature path* to be the following curve in $\mathbb{R}^d$:
$$\rho_K : [0, \infty] \to K \qquad \rho_K(r) = \mathsf{cg}(K + r\mathsf{B})$$

The path connects the centroid $\rho_K(0) = \mathsf{cg}(K)$ to the Steiner point $\rho_K(\infty)$. Our main result will exploit the fact that the coordinates of the curvature path are rational functions of bounded degree to produce a discretization.

**Lemma 5.1.** *Given $K \in \mathsf{Conv}_d$ and a discretization parameter $k$, there exists a set $D = \{p_0, p_1, \ldots, p_k\} \subset K$ such that for every $r$ there is a point $p_i \in D$ such that:*

$$|\langle \rho_K(r) - p_i, u\rangle| \le \tfrac{4d^3}{k} \cdot \mathsf{width}(K, u), \ \forall u \in \mathbb{S}^{d-1}.$$

## 5.2 Low-regret cutting-plane algorithms for weak separation oracles

In this section we use the discretization of the curvature path in Lemma 5.1 to construct a $\mathrm{poly}(d)$-regret cutting-plane algorithm that works against a weak separation oracle. Recall that a weak separation oracle is a separation oracle that fixes the direction of the output hyperplane in advance (up to sign). That is, at the beginning of round $t$ the oracle fixes some direction $v_t \in \mathbb{S}^{d-1}$ and returns either $v_t$ or $-v_t$ to the learner depending on the learner's choice of query point $q_t$.

**Theorem 5.2.** *The cutting-plane algorithm which chooses a random point from the discretization of the curvature path of $K_t$ into $d^4$ pieces achieves a total regret of $O(d^5 \log^2 d)$ against any weak separation oracle.*

## 5.3 List contextual recommendation

In this section, we consider the problem of list contextual recommendation. In this variant of contextual recommendation, we are allowed to offer a list of possible actions $L_t \subseteq \mathcal{X}_t$ and we measure regret against the best action in the list: $\mathsf{loss}_t = \langle w^*, x_t^* \rangle - \max_{x \in L_t} \langle w^*, x \rangle$.

Our main result is that if the list is allowed to be of size $O(d^4)$ then it is possible to achieve total regret $O(d^2 \log d)$.

The recommended list of actions will be computed as follows: given the knowledge set $K_t$, let $D$ be the discretization of the curvature path with parameter $k = 200d^4$ obtained in Lemma 5.1. Then for each $p_i \in D$ find an arbitrary $x_i \in \mathsf{BR}(p_i) := \arg\max_{x \in \mathcal{X}_t} \langle p_i, x \rangle$ and let $L_t = \{x_1, x_2, \ldots, x_k\}$.

**Theorem 5.3.** *There exists an algorithm which plays the list $L_t$ defined above and incurs a total regret of at most $O(d^2 \log d)$.*

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
