There is a very large body of work on recommender systems which employs a wide range of different techniques – for an overview, see the survey by Bobadilla et al. [5]. Our formulation in this paper is closest to treatments of recommender systems which formulate the problem as an online learning problem and attack it with tools such as contextual bandits or reinforcement learning. Some examples of such approaches can be seen in [17, 18, 23, 25, 26]. Similarly, there is a wide variety of work on online shortest path routing [3, 11, 12, 15, 24, 28] which also applies tools from online learning. One major difference between these works and the setting we study in our paper is that these settings often rely on some quantitative feedback regarding the quality of item recommended. In contrast, our paper only relies on qualitative feedback of the form "action $x$ is the best action this round" or "action $x$ is is at least as good as any action recommended".

One setting in the bandits literature that also possesses qualitative feedback is the setting of Duelling Bandits [27]. In this model, the learner can submit a pair of actions and the feedback is a noisy bit signalling which action is better. However, their notion of regret (essentially, the probability the best arm would be preferred over the arms chosen by the learner) significantly differs from the notion of

---

[3]The reason this type of algorithm does not work against strong separation oracles is that each point in this discretization could return a different direction $v_t$, in turn corresponding to a different value of $r$

229 regret we measure in our setting (the loss to the user by following our recommendations instead of
230 choosing the optimal actions).

231 Cutting-plane methods have a long and storied history in convex optimization. The very first efficient
232 algorithms for linear programming (based on the ellipsoid method [10, 14]). Since then, there has
233 been much progress in designing more efficient cutting-plane methods (e.g. [6]), but the focus remains
234 on the number of calls to the separating oracle or the total running time of the algorithm. We are not
235 aware of any work which studies cutting-plane methods under the notion of regret that we introduce
236 in Section 1.1.

237 Contextual search was first introduced in the form described in Section 2.2 in [16], where the authors
238 gave the first time-horizon-independent regret bound of $O(\text{poly}(d))$ for this problem (earlier work
239 by [20] and [9] indirectly implied bounds of $O(\text{poly}(d) \log T)$ for this problem). This was later
240 improved by [19] to a near-optimal $O(d \log d)$ regret bound. The algorithms of both [16, 19] rely
241 on techniques from integral geometry, and specifically on understanding the intrinsic volumes and
242 Steiner polynomial of the set of possible values for $w^*$. Some related geometric techniques have been
243 used in recent work on the convex body chasing problem[1, 7, 22]. To our knowledge, our paper is
244 the first paper to employ the fact that the curvature path $\text{cg}(K + r\text{B})$ is a bounded rational curve (and
245 thus can be efficiently discretized) in the development of algorithms.

## 246 2 Model and preliminaries

247 We begin by briefly reviewing the problems of contextual recommendation and designing low-regret
248 cutting plane algorithms. In all of the below problems, $\text{B} = \{w \in \mathbb{R}^d \mid \|w\|_2 \leq 1\}$ is the ball of
249 radius 1 (and generally, all vectors we consider will be bounded to lie in this ball).

250 **Contextual recommendation.** In *contextual recommendation* there is a hidden point $w^* \in \text{B}$.
251 Each round $t$ (for $T$ rounds) we are given a set of possible actions $\mathcal{X}_t \subseteq \text{B}$. If we choose
252 action $x_t \in \mathcal{X}_t$ we obtain reward $\langle x_t, w^* \rangle$ (but do not learn this value). Our feedback is
253 $x_t^* = \arg\max_{x \in \mathcal{X}_t} \langle x, w^* \rangle$, the identity of the best action[4]. Our goal is to minimize the total
254 expected regret $\mathbb{E}[\text{Reg}] = \mathbb{E}\left[\sum_{t=1}^T \langle x_t^* - x_t, w^* \rangle\right]$. Note that since the feedback is deterministic,
255 this expectation is only over the randomness of the learner's algorithm.

It will be useful to establish some additional notation for discussing algorithms for contextual
recommendation. We define the *knowledge set* $K_t$ to be the set of possible values for $w^*$ given the
knowledge we have obtained by round $t$. Note that the knowledge set $K_t$ is always convex, since
the feedback we receive each round (that $\langle x^*, w^* \rangle \geq \langle x, w^* \rangle$ for all $x \in \mathcal{X}_t$) can be written as an
intersection of several halfspaces (and the initial knowledge set $K_1 = \text{B}$ is convex). In fact, we can
say more. Given a $w \in K_t$, let

$$\text{BR}_t(w) = \arg\max_{x \in \mathcal{X}_t} \langle x, w \rangle$$

be the set of optimal actions in $\mathcal{X}_t$ if the hidden point was $w$. We can then partition $K_t$ into several
convex subregions based on the value of $\text{BR}_t(w)$; specifically, let

$$R_t(x) = \{w \in K_t | x \in \text{BR}_t(w)\}$$

256 be the region of $K_t$ where $x$ is the optimal action to play in response. Then:

257    1. Each $R_t(x)$ is a convex subset of $K_t$.

258    2. The regions $R_t(x)$ have disjoint interiors and partition $K_t$.

259    3. $K_{t+1}$ will equal the region $R_t(x^*)$ (where $x^* \in \text{BR}_t(w^*)$ is the optimal action returned as
260       feedback).

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

 \geq y_t\}$ be the halfspace defined by $H$, the two sets $(K_t \cap H^+) + r\mathsf{B}$ and $(K_t + r\mathsf{B}) \cap H^+$ are *not* equal. The volume of the first set represents the new value of our potential (i.e. $\mathsf{Vol}(K_{t+1} + r\mathsf{B})$), but it is the second set that has volume equal to half our current potential (i.e. $\frac{1}{2}\mathsf{Vol}(K_t + r\mathsf{B})$).

Luckily, our choice of $r$ allows us to relate these two quantities in a way so that our original argument works. Let $H$ divide $K$ into $K^+$ and $K^-$. Note that $\mathsf{Vol}(K^+ + r\mathsf{B}) + \mathsf{Vol}(K^- + r\mathsf{B}) = \mathsf{Vol}(K + r\mathsf{B}) + \mathsf{Vol}((K \cap H) + r\mathsf{B})$ (in particular, $K + r\mathsf{B}$ and $(K \cap H) + r\mathsf{B}$ are the union and intersection respectively of $K^+ + r\mathsf{B}$ and $K^- + r\mathsf{B}$). Since $\mathsf{Vol}(K^+ + r\mathsf{B}) = \mathsf{Vol}(K^- + r\mathsf{B})$, to bound $\mathsf{Vol}(K^+ + r\mathsf{B})/\mathsf{Vol}(K + r\mathsf{B})$ it suffices to bound $\mathsf{Vol}((K \cap H) + r\mathsf{B})$. We do so in the following lemma (which will also prove useful to us in later analysis).

**Lemma 1.** *Given $K \in \mathsf{Conv}_d$ and $u \in \mathbb{S}^{d-1}$, let $H$ be a hyperplane of the form $\{w \mid \langle w, u \rangle = b\}$ (for some $b \in \mathbb{R}$). Then:*

$$\mathsf{Vol}((K \cap H) + r\mathsf{B}) \leq \left( \frac{2rd}{\mathsf{width}(K; u)} \right) \cdot \mathsf{Vol}(K + r\mathsf{B})$$

*Proof.* Let $\overline{V} = \mathsf{Vol}_{d-1}((K + r\mathsf{B}) \cap H)$ be the volume of the $(d-1)$-dimensional cross-section of $K + r\mathsf{B}$ carved out by $H$. Note first that we can write any point in $(K \cap H) + r\mathsf{B}$ in the form $w + \lambda u$, where $w \in (K + r\mathsf{B}) \cap H$ and $\lambda \in [-r, r]$. It follows that

$$\mathsf{Vol}((K \cap H) + r\mathsf{B}) \leq 2r\overline{V}. \tag{3}$$

We will now bound $\overline{V}$. Let $\overline{K} = (K + r\mathsf{B}) \cap H$. Let $p^+$ be the point in $K + r\mathsf{B}$ maximizing $\langle u, p \rangle$, and let $p^-$ be the point in $K + r\mathsf{B}$ minimizing $\langle u, p \rangle$ (so $p^-$ and $p^+$ certify the width). Consider the cones $C^-$ and $C^+$ formed by taking the convex hull $\mathsf{Conv}(p^-, \overline{K})$ and $\mathsf{Conv}(p^+, \overline{K})$ respectively. $C^-$ and $C^+$ are disjoint and contained within $K + r\mathsf{B}$, so

$$\mathsf{Vol}(C^-) + \mathsf{Vol}(C^+) \leq \mathsf{Vol}(K + r\mathsf{B}).$$

But now note that by the formula for the volume of a cone,

$$\mathsf{Vol}(C^-) + \mathsf{Vol}(C^+) = \frac{1}{d} \cdot \mathsf{width}(K + r\mathsf{B}; u) \cdot \mathsf{Vol}_{d-1}(\overline{K}) \geq \frac{\mathsf{width}(K; u)}{d} \cdot \overline{V}.$$

It follows that

$$\overline{V} \le \frac{d}{\mathsf{width}(K; u)} \mathsf{Vol}(K + r\mathsf{B}). \tag{4}$$

Substituting this into (3), we arrive at the theorem statement. $\qquad\square$

This lemma allows us to conclude our analysis of the contextual search algorithm. In particular, since we have chosen $r \approx \mathsf{width}(K, v_t)/10d$, by applying this lemma we can see that in our analysis of contextual search, $\mathsf{Vol}((K \cap H) + r\mathsf{B}) \le 0.2\mathsf{Vol}(K + r\mathsf{B})$, from which it follows that $\mathsf{Vol}(K^+ + r\mathsf{B})/\mathsf{Vol}(K + r\mathsf{B}) \le 0.6$.

## 3 From Cutting-Plane Algorithms to Contextual Recommendation

We begin by proving a reduction from designing low-regret cutting plane algorithms to contextual recommendation. Specifically, we will show that given a regret $\rho$ cutting-plane algorithm, we can use it to construct an $O(\rho)$-regret algorithm for contextual recommendation.

Note that while these two problems are similar in many ways (e.g. they both involve searching for an unknown point $w^*$), they are not completely identical. Among other things, the formulation of regret although similar is qualitatively different between the two problems (i.e. between expressions (1) and (2)). In particular, in contextual recommendation, the regret each round is $\langle x_t^* - x_t, w^* \rangle$, whereas for cutting-plane algorithms, the regret is given by $\langle w^* - p_t, v_t \rangle$. Nonetheless, we will be able to relate these two notions of regret by considering a separation oracle that always returns a halfspace in the direction of $x_t^* - x_t$. We present this reduction below.

**Theorem 4.** *Given a low-regret cutting-plane algorithm $\mathcal{A}$ with regret $\rho$, we can construct an $O(\rho)$-regret algorithm for contextual recommendation.*

*Proof.* We will simultaneously run an instance of $\mathcal{A}$ with the same hidden vector $w^*$. Each round we will ask $\mathcal{A}$ for its query $p_t$ to the separation oracle. We will then compute a $x_t \in \mathsf{BR}_t(p_t)$ (recall that $\mathsf{BR}_t(w)$ is the optimal action to play if $w$ is the true hidden vector) and submit $x_t$ as our action for this round of contextual recommendation. We then receive feedback $x_t^* \in \mathsf{BR}_t(w^*)$. Consider the following two cases:

**Case 1:** If $x_t^* = x_t$, then our contextual recommendation algorithm incurs zero regret since we successfully chose the optimal point. In this case we ignore this round for $\mathcal{A}$ (i.e. we reset its state to its state at the beginning of round $t$).

**Case 2:** If $x_t^* \ne x_t$, let $v_t = (x_t^* - x_t)/\|x_t^* - x_t\|$. We will return $v_t$ to $\mathcal{A}$ as the separation oracle's answer to query $p_t$. Note that this is a valid answer, since

$$\langle w^* - p_t, v_t \rangle = \frac{1}{\|x_t^* - x_t\|} \left( \langle w^*, x_t^* - x_t \rangle + \langle p_t, x_t - x_t^* \rangle \right) \ge \frac{1}{\|x_t^* - x_t\|} \langle w^*, x_t^* - x_t \rangle. \tag{5}$$

Here the final inequality holds since (by the definition of $\mathsf{BR}_t(p_t)$) $\langle p_t, x_t \rangle \ge \langle p_t, x \rangle$ for any $x \in \mathcal{X}_t$. The RHS of (5) is in turn larger than zero, since $\langle w^*, x_t^* \rangle \ge \langle w^*, x \rangle$ for any $x \in \mathcal{X}_t$ (and thus this is a valid answer to the separation oracle). Moreover, note that the regret we incur under contextual recommendation is exactly $\langle w^*, x_t^* - x_t \rangle$, so by rearranging equation (5), we have that:

$$\langle w^*, x_t^* - x_t \rangle \le \|x_t^* - x_t\| \langle w^* - p_t, v_t \rangle \le 2\langle w^* - p_t, v_t \rangle.$$

It follows that the total regret of our algorithm for contextual recommendation is at most twice that of $\mathcal{A}$. Our regret is thus bounded above by $2\rho$, as desired.

$\qquad\square$

Note that the reduction in Theorem 4 is efficient as long as we have an efficient method for optimizing a linear function over $\mathcal{X}_t$ (i.e. for computing $\mathsf{BR}_t(w)$). In particular, this means that this reduction

can be practical even in settings where $\mathcal{X}_t$ may be combinatorially large (e.g. the set of $s$-$t$ paths in some graph).

Note also that this reduction *does not* work if $\mathcal{A}$ is only low-regret against weak separation oracles. This is since the direction $v_t$ we choose does depend non-trivially on the point $p_t$ (in particular, we choose $x_t \in \mathsf{BR}_t(p_t)$). Later in Section 5.3, we will see how to use ideas from designing cutting-plane methods for weak separation oracles to construct low-regret algorithms for *list* contextual recommendation – however we do not have a black-box reduction in that case, and our construction will be more involved.

# 4 Designing Low-Regret Cutting-Plane Algorithms

In this section we will describe how to construct low-regret cutting-plane algorithms for strong separation oracles.

## 4.1 An $\exp(O(d \log d))$-regret cutting-plane algorithm

We begin with a quick proof that always querying the center of the John ellipsoid of $K_t$ leads to a $\exp(O(d \log d))$-regret cutting-plane algorithm. Interestingly, although this corresponds to the classical ellipsoid algorithm, our analysis will instead proceed along the lines of the analysis of the contextual search algorithm summarized in Section 2.2.

We will need the following lemma.

**Lemma 2.** *Let $K \in \mathsf{Conv}_d$ be an arbitrary convex set and let $r \geq 0$. Let $E$ be the John ellipsoid of $K$, and let $H$ be a hyperplane that passes through the center of $E$, dividing $K$ into two regions $K^+$ and $K^-$. Then*

$$\mathsf{Vol}(K^+ + r\mathsf{B}) \leq \left(1 - \frac{1}{10d^d}\right)\left(\mathsf{Vol}(K^+ + r\mathsf{B}) + \mathsf{Vol}(K^- + r\mathsf{B})\right)$$

*Proof.* Let $H$ divide $E$ into the two regions $E^+$ and $E^-$ analogously to how it divides $K$ into $K^+$ and $K^-$. Note that since $E \subseteq K \subseteq dE$ (translating $K$ so that $E$ is centered at the origin), we can write:

$$\frac{\mathsf{Vol}(K^- + r\mathsf{B})}{\mathsf{Vol}(K + r\mathsf{B})} \geq \frac{\mathsf{Vol}(E^- + r\mathsf{B})}{\mathsf{Vol}(dE + r\mathsf{B})} \geq \frac{0.5 \cdot \mathsf{Vol}(E + r\mathsf{B})}{\mathsf{Vol}(dE + r\mathsf{B})} \geq \frac{1}{2d^d}\frac{\mathsf{Vol}(E + r\mathsf{B})}{\mathsf{Vol}(E + (r/d)\mathsf{B})} \geq \frac{1}{2d^d}. \quad (6)$$

On the other hand, by monotonicity we also have that

$$\frac{\mathsf{Vol}(K^+ + r\mathsf{B})}{\mathsf{Vol}(K + r\mathsf{B})} \leq 1.$$

It follows that

$$\mathsf{Vol}(K^+ + r\mathsf{B})/\mathsf{Vol}(K^- + r\mathsf{B}) \leq 2d^d.$$

The conclusion then follows since

$$2d^d \leq \left(1 - \frac{1}{10d^d}\right)(2d^d + 1).$$

$\square$

We can now modify the analysis of contextual search to make use of Lemma 2. In particular, we will show that for each round $t$, there's some $r$ (roughly proportional to the current width) where $\mathsf{Vol}(K_t + rB)$ decreases by a multiplicative factor of $(1 - d^{-O(d)})$.

**Theorem 5.** *The cutting-plane algorithm which always queries the center of the John ellipsoid of $K_t$ incurs $\exp(O(d \log d))$ regret.*

*Proof.* Fix a round $t$, and let $K = K_t$ be the knowledge set at time $t$. Let $E$ be the John ellipsoid of $K$ and let $p_t$ be the center of $E$. When we query the separation oracle with $p_t$, we get a hyperplane $H$ (defined by $v_t$) that passes through $p_t$ and divides $K$ into $K^+ = K_{t+1}$ and $K^- = K \setminus K_{t+1}$.

By Lemma 2, for any $r \geq 0$, we have that

$$\mathsf{Vol}(K^+ + r\mathsf{B}) \leq \left(1 - \frac{1}{10d^d}\right)\left(\mathsf{Vol}(K^+ + r\mathsf{B}) + \mathsf{Vol}(K^- + r\mathsf{B})\right)$$

Note that (as in Section 2.2), $\mathsf{Vol}(K^+ + r\mathsf{B}) + \mathsf{Vol}(K^- + r\mathsf{B}) = \mathsf{Vol}(K + r\mathsf{B}) + \mathsf{Vol}((K \cap H) + r\mathsf{B})$. By Lemma 1, we have that

$$\mathsf{Vol}((K \cap H) + r\mathsf{B}) \leq \frac{2rd}{\mathsf{width}(K; v_t)} \cdot \mathsf{Vol}(K + r\mathsf{B}),$$

and thus that

$$\mathsf{Vol}(K^+ + r\mathsf{B}) \leq \left(1 - \frac{1}{10d^d}\right)\left(1 + \frac{2dr}{\mathsf{width}(K; v_t)}\right)\mathsf{Vol}(K + r\mathsf{B})$$

In particular, if we choose $r \leq \mathsf{width}(K; v_t)/(100d^{d+1})$, then

$$\mathsf{Vol}(K^+ + r\mathsf{B}) \leq \left(1 - \frac{1}{20d^d}\right)\mathsf{Vol}(K + r\mathsf{B}).$$

The analysis now proceeds as follows. In each round, let $r = 2^{\lfloor \lg(\mathsf{width}(K; v_t)/100d^{d+1})\rfloor}$ be the largest power of 2 smaller than $w/(100d^{d+1})$. Any specific $r$ can occur in at most

$$\frac{\log(\mathsf{Vol}(K_0 + r\mathsf{B})/\mathsf{Vol}(K_T + r\mathsf{B}))}{\log\left(1 - \frac{1}{20d^d}\right)}$$

rounds. This in turn is at most

$$\frac{\log(\mathsf{Vol}(2\mathsf{B})/\mathsf{Vol}(r\mathsf{B}))}{1/(20d^d)} \leq 20d^{d+1}\log(2/r)$$

rounds, and in each such round the regret that round is at most $\mathsf{width}(K; v_t) \leq 200d^{d+1}r$. The total regret from such rounds is therefore at most

$$20d^{d+1}\log(2/r) \cdot 200d^{d+1}r = O(d^{2(d+1)}r\log(2/r)).$$

Now, by our discretization, $r$ is a power of two less than 1. Note that $\sum_{i=0}^{\infty} 2^{-i}\log(2/2^{-i}) = O\left(\sum_{i=0}^{\infty} 2^{-i}i\right) = O(1)$. It follows that the total regret over all rounds is at most $O(d^{2(d+1)}) = \exp(O(d \log d))$, as desired. $\square$

The remaining algorithms we study will generally query the center-of-gravity of some convex set, as opposed to the center of the John ellipsoid. This leads to the following natural question: what is the regret of the cutting-plane algorithm which always queries the center-of-gravity of $K_t$?

Kannan, Lovasz, and Simonovits (Theorem 4.1 of [13]) show that it is possible to choose an ellipsoid $E$ satisfying $E \subseteq K \subseteq dE$ such that $E$ is centered at $\mathsf{cg}(K)$, so our proof of Theorem 5 shows that this algorithm is also an $\exp(O(d \log d))$ algorithm. However, for both this algorithm and the ellipsoid algorithm of Theorem 5, we have no non-trivial lower bound on the regret. It is an interesting open question to understand what regret these algorithms actually obtain (for example, do either of these algorithms achieve $\mathrm{poly}(d)$ regret?).

## 4.2 An $O(d \log T)$-regret cutting-plane algorithm

We will now show how to obtain an $O(d \log T)$-regret cutting plane algorithm. Our algorithm will simply query the center-of-gravity of $K_t + \frac{1}{T}\mathsf{B}$ each round. The advantage of doing this is that we will only need to examine one scale of the contextual search potential (namely the value of $\mathsf{Vol}(K_t + \frac{1}{T}\mathsf{B})$). The following geometric lemma shows that, as long as the width of the $K_t$ is long enough, this potential decreases by a constant fraction each step.

**Lemma 3.** *Given $K \in \mathsf{Conv}_d$, $u \in \mathbb{S}^{d-1}$ and $b, r \in \mathbb{R}$ (with $r \geq 0$), let:*

- *$c = \mathsf{cg}(K + r\mathsf{B})$ be the center-of-gravity of $K + r\mathsf{B}$,*
- *$H^+(b) = \{\langle u, x - c \rangle \geq -b\}$ be a half-space induced by a hyperplane in the direction $u$ passing within distance $b$ of the point $c$, and*
- *$K^+ = K \cap H^+(b)$ be the intersection of $K$ with this half-space.*

*If $r, |b| \leq \mathsf{width}(K, u)/(16ed)$ then*

$$\mathsf{Vol}(K^+ + r\mathsf{B}) \leq 0.9 \cdot \mathsf{Vol}(K + r\mathsf{B}).$$

*Proof.* Observe that $K^+ + rB \subseteq (K + rB) \cap H^+(b+r)$. If we define $H^-(b+r) = \{x \in \mathbb{R}^d; \langle u, x - c \rangle \leq -(b+r)\}$ then:

$$\mathsf{Vol}(K^+ + rB) \geq \mathsf{Vol}(K + rB) - \mathsf{Vol}((K + rB) \cap H^-(b+r)).$$

By Theorem 2 (Approximate Grunbaum) we have:

$$\frac{\mathsf{Vol}((K + rB) \cap H^-(b+r))}{\mathsf{Vol}(K + rB)} \geq \frac{1}{e} - \frac{2(d+1)}{\mathsf{width}(K; u)} \cdot \frac{2\mathsf{width}(K, u)}{16ed} \geq \frac{1}{2e} \geq 0.1$$

$\square$

We can now prove that the above algorithm achieves $O(d \log T)$ regret.

**Theorem 6.** *The cutting-plane algorithm which queries the point $p_t = \mathsf{cg}\left(K_t + \frac{1}{T}\mathsf{B}\right)$ incurs $O(d \log T)$ regret.*

*Proof.* We will begin by showing that if we incur more than $50d/T$ regret in a given round, we reduce the value of $\mathsf{Vol}(K_t + \frac{1}{T}\mathsf{B})$ by a constant factor. Since $\mathsf{Vol}(K_t + \frac{1}{T}\mathsf{B})$ is bounded below by $\mathsf{Vol}(\frac{1}{T}\mathsf{B})$, this will allow us to bound the number of times we incur a large amount of regret.

Consider a fixed round $t$ of this algorithm. Let $K_t$ be the knowledge set at time $t$. When we query the separation-oracle point $p_t = \mathsf{cg}(K + \frac{1}{T}\mathsf{B})$, we obtain a half-space $H^+ = \{w \in \mathbb{R}^d; \langle w - p, v_t \rangle \geq 0\}$ passing through $p_t$ which contains $w^*$. We update $K_{t+1} = K_t \cap H^+$

The regret in round $t$ is bounded by $\mathsf{width}(K_t, v_t)$. If the width is at least $50d/T$ we can then apply Lemma 3 with $b = 0$ and $r = 1/T$ to conclude that:

$$\mathsf{Vol}\left(K_{t+1} + \frac{1}{T}\mathsf{B}\right) \leq 0.9 \cdot \mathsf{Vol}\left(K_t + \frac{1}{T}\mathsf{B}\right). \tag{7}$$

Now, in each round where $\mathsf{width}(K_t, v_t) < 50d/T$, we incur at most $50d/T$ regret, so in total we incur at most $T \cdot (50d/T) = 50d$ regret from such rounds. On the other hand, in other rounds we may incur up to $\|w^* - p_t\| \leq 2$ regret per round. However, note that $\mathsf{Vol}(K_1 + \frac{1}{T}\mathsf{B}) = \mathsf{Vol}((1 + \frac{1}{T})\mathsf{B}) \leq 2^d\mathsf{Vol}(B)$, whereas for any $t$, $\mathsf{Vol}(K_t + \frac{1}{T}\mathsf{B}) \geq \mathsf{Vol}(\frac{1}{T}\mathsf{B}) = T^{-d}\kappa_d$. Since in each such round we shrink this quantity by at least a factor of $0.9$, it follows that the total number of such rounds is at most $O(\log(2T^d)) = O(d \log T)$. It follows that the total regret from such rounds is at most $O(d \log T)$, and thus the overall regret of this algorithm is at most $O(d \log T)$. $\square$

# 5 List contextual recommendation, weak separation oracles, and the curvature path

In this section, we present two algorithms: 1. a $\mathrm{poly}(d)$ expected regret cutting-plane algorithm for weak separation oracles, and 2. an $O(d^2 \log d)$ regret algorithm for list contextual recommendation with list size $L = \mathrm{poly}(d)$.

The unifying feature of both algorithms is that they both involve analyzing a geometric object we call the *curvature path* of a convex body. The *curvature path* of $K$ is a bounded-degree rational curve contained within $K$ that connects the center-of-gravity $\mathsf{cg}(K)$ with the Steiner point ($\lim_{r\to\infty} \mathsf{cg}(K + r\mathsf{B})$) of $K$.

In Section 5.1 we formally define the curvature path and demonstrate how to bound its length. In Section 5.2, we show that randomly querying a point on a discretization of the curvature path leads to a $\mathsf{poly}(d)$ regret cutting-plane algorithm for weak separation oracles. Finally, in Section 5.1, we show how to transform a discretization of the curvature path of the knowledge set into a list of actions for list contextual recommendation, obtaining a low regret algorithm.

## 5.1 The curvature path

An important fact (driving some of the recent results in contextual search, e.g. [16]) is the fact that the volume $\mathsf{Vol}(K + r\mathsf{B})$ is a $d$-dimensional polynomial in $r$. This fact is known as the Steiner formula:

$$\mathsf{Vol}(K + r\mathsf{B}) = \sum_{i=0}^{d} V_{d-i}(K)\kappa_i r^i \tag{8}$$

After normalization by the volume of the unit ball, the coefficients of this polynomial correspond to the *intrinsic volumes* of $K$. The intrinsic volumes are a family of $d + 1$ functionals $V_i : \mathsf{Conv}_d \to \mathbb{R}_+$ for $i = 0, 1, \ldots, d$ that associate for each convex $K \in \mathsf{Conv}_d$ a non-negative value. Some of these functionals have natural interpretations: $V_d(K)$ is the standard volume $\mathsf{Vol}(K)$, $V_{d-1}(K)$ is the surface area, $V_1(K)$ is the average width and $V_0(K)$ is 1 whenever $K$ is non-empty and 0 otherwise.

There is an analogue of the Steiner formula for the centroid of $K + r\mathsf{B}$, showing that it admits a description as a vector-valued rational function. More precisely, there exist $d + 1$ functions $c_i : \mathsf{Conv}_d \to \mathbb{R}^d$ for $0 \le i \le d$ such that:

$$\mathsf{cg}(K + r\mathsf{B}) = \frac{\sum_{i=0}^{d} V_{d-i}(K)\kappa_i r^i \cdot c_i(K)}{\sum_{i=0}^{d} V_{d-i}(K)\kappa_i r^i} \tag{9}$$

The point $c_0(K) \in K$ corresponds to the usual centroid $\mathsf{cg}(K)$ and $c_d(K)$ corresponds to the Steiner point. The functionals $c_i$ are called *curvature centroids* since they can be computing by integrating a certain curvature measures associated with a convex body (a la Gauss-Bonnet). We refer to Section 5.4 in Schneider [21] for a more thorough discussion discussion. For our purposes, however, the only important fact will be that each curvature centroid $c_i(K)$ is guaranteed to lie within $K$ (note that this is not at all obvious from their definition).

Motivated by this, given a convex body $K \subseteq \mathbb{R}^d$ we define its *curvature path* to be the following curve in $\mathbb{R}^d$:

$$\rho_K : [0, \infty] \to K \qquad \rho_K(r) = \mathsf{cg}(K + r\mathsf{B})$$

The path connects the centroid $\rho_K(0) = \mathsf{cg}(K)$ to the Steiner point $\rho_K(\infty)$. Our main result will exploit the fact that the coordinates of the curvature path are rational functions of bounded degree to produce a discretization. We start by bounding the length of the path. For reasons that will become clear, it will be more convenient to bound its length when transformed by the linear map in John's Theorem.

**Lemma 4.** *Let $K \in \mathsf{Conv}_d \setminus \{\emptyset\}$, and let $A$ be a linear transformation as in (John's) Theorem 3. Then the length of the path $\{A\rho_K(r); r \in [0, \infty]\}$ is at most $4d^3$.*

*Proof.* The length of a path is the integral of the $\ell_2$-norm of its derivative. We will bound the $\ell_2$ norm by the $\ell_1$ norm and then analyze each of its components.

$$\mathsf{length}(A\rho_K) = \int_0^\infty \|A\rho_K'(r)\|_2 dr \le \int_0^\infty \|A\rho_K'(r)\|_1 dr = \sum_{i=1}^{d} \int_0^\infty |(A\rho_K'(r))_i| dr \tag{10}$$

where $(A\rho_K'(r))_i$ is the $i$-th component of the vector $A\rho_K'(r)$. By equation (9), we know that there are degree-$d$ polynomials $p(r)$ and $q(r)$ such that $(A\rho_K'(r))_i = p(r)/q(r)$ where $q(r) > 0$ for all

$r \geq 0$. Hence we can write its derivative as: $(A\rho'_K(r))_i = (p'(r)q(r) - p(r)q'(r))/(q(r)^2)$ which can be re-written as $h(r)/q(r)^2$ for a polynomial $h(r)$ of degree at most $2d - 1$. Now a polynomial of degree at most $k$ can change signs at most $k$ times. So we can partition $[0, \infty]$ into at most $2d$ intervals $I_1, \ldots, I_{2d}$ (some possibly empty) such that the sign of $(A\rho'_K(r))_i$ is the same within each region (treating zeros arbitrarily). If $I_j = [a_j, b_j]$, we can then write:

$$\int_0^\infty |(A\rho'_K(r))_i| dr = \sum_{j=1}^{2d} \int_{a_j}^{b_j} |(A\rho'_K(r))_i| = \sum_{i=1}^{2d} |(A\rho_K(b_j))_i - (A\rho_K(a_j))_i| \leq 4d^2 \quad (11)$$

where the last step follows from John's theorem. Since $A(\rho_K)$ is in $A(K)$ which is contained in a ball of radius $d$, the distance between the $i$-coordinate of two points is at most $2d$. Equations (10) and (11) together imply the statement of the lemma. $\qquad\square$

**Lemma 5.** *Given* $K \in \mathsf{Conv}_d$ *and a discretization parameter* $k$, *there exists a set* $D = \{p_0, p_1, \ldots, p_k\} \subset K$ *such that for every* $r$ *there is a point* $p_i \in D$ *such that:*

$$|\langle \rho_K(r) - p_i, u \rangle| \leq \frac{4d^3}{k} \cdot \mathsf{width}(K, u), \; \forall u \in \mathbb{S}^{d-1}.$$

*Proof.* Discretize the path $A\rho_k$ into $k$ pieces of equal length and let $Ap_0, Ap_1, \ldots, Ap_k$ correspond to the endpoints. Let $D = \{p_0, p_1, \ldots, p_k\}$. We know by Lemma 4 that for any $p = \rho_K(r)$, there exists a $p_i \in D$ such that: $\|Ap_i - Ap\|_2 \leq 4d^3/k$.

Now, for each unit vector $u \in \mathbb{S}^{d-1}$, we have:

$$|\langle u, p_i - p \rangle| \leq \langle A^{-T}u, A(p_i - p) \rangle \leq \|A^{-T}u\| \cdot \|A(p_i - p)\| \leq \|A^{-T}u\| \cdot 4d^3/k$$

Finally, we argue that $\|A^{-T}u\| \leq \mathsf{width}(K; u)$. Let $v = (A^{-T}u)/\|A^{-T}u\|$ and take $x, y \in K$ that certify the width of $K$ in direction $u$:

$$\mathsf{width}(K, u) = \langle u, x - y \rangle = \langle A^{-T}u, Ax - Ay \rangle = \|A^{-T}u\| \cdot \langle v, Ax - Ay \rangle$$

Finally note that $Ax$ and $Ay$ are respectively the maximizer and minimizer of $\langle v, z \rangle$ for $z \in A(K)$ since: $\max_{z \in A(K)} \langle v, z \rangle = \max_{x \in K} \langle v, Ax \rangle = \max_{x \in K} \langle A^T v, x \rangle = \max_{x \in K} \langle u, x \rangle / \|A^{-T}u\|$. This implies that $\langle v, Ax - Ay \rangle = \mathsf{width}(A(K), v) \geq 1$ by John's Theorem since $q + \mathsf{B} \subseteq A(K)$. This completes the proof. $\qquad\square$

## 5.2 Low-regret cutting-plane algorithms for weak separation oracles

In this section we show how to use the discretization of the curvature path in Lemma 5 to construct a $\mathsf{poly}(d)$-regret cutting-plane algorithm that works against a weak separation oracle.

Recall that a weak separation oracle is a separation oracle that fixes the direction of the output hyperplane in advance (up to sign). That is, at the beginning of round $t$ the oracle fixes some direction $v_t \in \mathbb{S}^{d-1}$ and returns either $v_t$ or $-v_t$ to the learner depending on the learner's choice of query point $q_t$.

One advantage of working with a weak separation oracle is that the width $\mathsf{width}(K_t; v_t)$ of the knowledge set in the direction $v_t$ is fixed and independent of the query point $p_t$ of the learner. This means that if we can guess the width, we can run essentially the standard contextual search algorithm (of Section 2.2) by querying any point $p_t$ that lies on the hyperplane which decreases the potential corresponding to this width by a constant factor. One good way to guess the width turns out to choose a random point belonging to a suitably fine discretization of the curvature path.

**Theorem 7.** *The cutting-plane algorithm which chooses a random point from the discretization of the curvature path of* $K_t$ *into* $d^4$ *pieces achieves a total regret of* $O(d^5 \log^2 d)$ *against any weak separation oracle.*

*Proof.* Consider a fixed round $t$. Let $v_t$ be the direction fixed by the weak separation-oracle and let $\omega = \mathsf{width}(K_t; v_t)$. Let $r = 2^{\lceil \lg(\omega/16ed) \rceil}$ (rounding $\omega/16ed$ to the nearest power of two).

If we could choose the point $p_t = \rho_{K_t}(r) = \mathsf{cg}(K_t + r\mathsf{B})$, then by Lemma 3, any separating hyperplane through $p_t$ would decrease this potential by a constant factor. However, we do not know

r. Instead, we will choose a random point from the discretization $D$ of the curvature path of $K_t$ into $O(d^4)$ pieces, and argue that by Lemma 5 one of these points will be close enough to $\rho_{K_t}(r)$ to make the argument go through.

Formally, let $D$ be the discretization of $\rho_{K_t}$ into $64ed^4$ pieces as per Lemma 5. By Lemma 5, there then exists a point $p_i \in D$ that satisfies

$$|\langle \rho_K(r) - p_i, v_t \rangle| \leq \frac{1}{16ed} \cdot \mathsf{width}(K_t; v_t). \tag{12}$$

Let $H$ be a hyperplane through $p_i$ in the direction $v_t$ (i.e. $H = \{\langle w - p_i, v_t \rangle = 0\}$), and let $H$ divide $K_t$ into the two regions $K^+$ and $K^-$. By Lemma 3 (with $b = \langle \rho_K(r) - p_i, v_t \rangle$), since (12) holds, we have that

$$\mathsf{Vol}(K^+ + r\mathsf{B}) \leq 0.9 \cdot \mathsf{Vol}(K + r\mathsf{B}). \tag{13}$$

Now, consider the algorithm which queries a random point in $D$. With probability $1/|D| = \Omega(d^{-4})$, equation (13) holds. Otherwise, it is still true that $\mathsf{Vol}(K^+ + r\mathsf{B}) \leq \cdot \mathsf{Vol}(K + r\mathsf{B})$. Therefore in expectation,

$$\mathbb{E}[\mathsf{Vol}(K_{t+1} + r\mathsf{B})] \leq \left(1 - \Omega(d^{-4})\right) \mathbb{E}[\mathsf{Vol}(K_t + r\mathsf{B})].$$

In particular, the total expected number of rounds we can have where $r = 2^{-i}$ is at most $di/\log(1/(1 - \Omega(d^{-4}))) = O(id^5)$. In such a round, our maximum possible loss is at most $\mathsf{width}(K_t; v_t) \leq \min(20dr, 2)$. Summing over all $i$ from 0 to $\infty$, we arrive at a total regret bound of

$$\sum_{i=0}^{\infty} O(id^5 \min(d2^{-i}, 1)) = \sum_{i=0}^{\log d} O(id^5) + d^6 \sum_{i=\log d}^{\infty} O(i2^{-i}) = O(d^5 \log^2 d).$$

$\square$

## 5.3 List contextual recommendation

In this section, we consider the problem of list contextual recommendation. In this variant of contextual recommendation, we are allowed to offer a list of possible actions $L_t \subseteq \mathcal{X}_t$ and we measure regret against the best action in the list:

$$\mathsf{loss}_t = \langle w^*, x_t^* \rangle - \max_{x \in L_t} \langle w^*, x \rangle.$$

Our main result is that if the list is allowed to be of size $O(d^4)$ then it is possible to achieve total regret $O(d^2 \log d)$.

The recommended list of actions will be computed as follows: given the knowledge set $K_t$, let $D$ be the discretization of the curvature path with parameter $k = 200d^4$ obtained in Lemma 5. Then for each $p_i \in D$ find an arbitrary $x_i \in \mathsf{BR}(p_i) := \arg\max_{x \in \mathcal{X}_t} \langle p_i, x \rangle$ and let $L_t = \{x_1, x_2, \ldots, x_k\}$.

**Theorem 8.** *There exists an algorithm which plays the list $L_t$ defined above and incurs a total regret of at most $O(d^2 \log d)$.*

*Proof.* The overall structure of the proof will be as follows: we will show that for each integer $j \geq 0$, the algorithm can incur loss between $100d \cdot 2^{-j}$ and $200d \cdot 2^{-j}$ at most $O(jd)$ times. Hence the total loss of the algorithm can be bounded by $\sum_{j=1}^{\infty} O(jd) \cdot 2^{-j} d \leq O(d^2 \log d)$.

*Potential function:* This will be done via a potential function argument. As usual, we will keep track of knowledge $K_t$ which corresponds to all possible values of $w$ that are consistent with the observations seen so far. $K_1 = \mathsf{B}$ and:

$$K_{t+1} = K_t \cap \left[\cap_{i \in L_t}\{w \in \mathbb{R}^d; \langle x^* - x, w \rangle \geq 0\}\right]$$

Associated with $K_t$ we will keep track of a family of potential functions:
$$\Phi_t^j = \mathsf{Vol}(K_t + 2^{-j}\mathsf{B})$$

Since $K_1 \supseteq K_2 \supseteq K_3 \supseteq ...$ the potentials will be non-increasing: $\Phi_1^j \geq \Phi_2^j \geq \Phi_3^j \geq ....$ One other important property is that the potential functions are lower bounded:
$$\Phi_j^t \geq \mathsf{Vol}(2^{-j}\mathsf{B}) = 2^{-jd}\mathsf{Vol}(\mathsf{B}) \tag{14}$$

We will argue that if we can bound the loss at any given step $t$ by $200 \cdot 2^{-j}d$, then $\Phi_{t+1}^j \leq 0.9 \cdot \Phi_t^j$. Because of the lower bound in equation 14, this can happen at most
$$O\left(\log\left(\frac{\Phi_j^1}{2^{-jd}\mathsf{Vol}(B)}\right)\right) = O\left(\log\left(\frac{(1+2^{-j})^d\mathsf{Vol}(\mathsf{B})}{2^{-jd}\mathsf{Vol}(B)}\right)\right) \leq O(jd)$$

*Bounding the loss:* We start by bounding the loss and depending on the loss we will show a constant decrease in a corresponding potential function. Let
$$x^* \in \arg\max_{x \in \mathcal{X}_t}\langle w^*, x\rangle$$

If $x^*$ is in the convex hull of $L_t$ then there must some of the points in $x_i \in L_t$ that is also optimal, in which case the algorithm incurs zero loss in this round and we can ignore it. Otherwise, we can assume that $x^*$ is not in the convex hull of $L_t$.

In that case, define for each $x_i \in L_t$ the vector:
$$v_i = \frac{x^* - x_i}{\|x^* - x_i\|_2}$$

Consider the index $i$ that minimizes $\mathsf{width}(K; v_i)$ and use this point to bound the loss:
$$\mathsf{loss}_t = \min_{x \in L_t}\langle w^*, x^* - x\rangle \leq \langle w^*, x^* - x_i\rangle \leq \langle w^* - p_i, x^* - x_i\rangle$$
$$= \langle w^* - p_i, v_i\rangle \cdot \|x^* - x_i\| \leq 2\langle w^* - p_i, v_i\rangle \leq 2\mathsf{width}(K, v_i)$$

The second inequality above follows from the definition of $x_i$ since $x_i \in \arg\max_{x \in \mathcal{X}_t}\langle p_i, x\rangle$ it follows that $\langle p_i, x_i - x^*\rangle \geq 0$.

*Charging the loss to the potential* We will now charge this loss to the potential. For that we first define an index $j$ such that:
$$j = -\left\lceil\frac{\mathsf{width}(K, v_i)}{100d}\right\rceil$$

With this definition we have:
$$\mathsf{loss}_t \leq 2\mathsf{width}(K, v_i) \leq 200d2^{-j}$$

Our final step is to show that the potential $\Phi_t^j$ decreases by a constant factor. For that we will use a combination of the discretization in Theorem 5 and the volume reduction guarantee in Lemma 3.

First consider the point:
$$g_i = \mathsf{cg}(K + 2^{-j}\mathsf{B})$$
Since it is on the curvature path, there is a discretized point $p_\ell \in D$ such that:
$$|\langle v_\ell, g_i - p_\ell\rangle| \leq \mathsf{width}(K, v_\ell)/(50d)$$
Together with the facts that $\langle w^*, v_\ell\rangle \geq 0$ and $\langle p_\ell, v_\ell\rangle \leq 0$ we obtain that:
$$\langle w^* - g_i, v_\ell\rangle = \langle w^* - p_\ell, v_\ell\rangle + \langle p_\ell - g_i, v_\ell\rangle \geq -\mathsf{width}(K, v_\ell)/(50d)$$
This in particular implies that:
$$K_{t+1} \subseteq \tilde{K}_{t+1} := K_t \cap \{w \in \mathbb{R}^d; \langle w - g_i, v_\ell\rangle \geq -\mathsf{width}(K, v_\ell)/(50d)\}$$
We are now in the position of applying Lemma 3 with $r = 2^{-j}$. Note that
$$r = 2^{-j} \leq \frac{\mathsf{width}(K, v_i)}{50d} \leq \frac{\mathsf{width}(K, v_\ell)}{50d}$$
where the last inequality follows from the choice of the index $i$ as the one minimizing $\mathsf{width}(K, v_i)$. Applying the Theorem, we obtain that:
$$\mathsf{Vol}(K_{t+1} + 2^{-j}\mathsf{B}) \leq \mathsf{Vol}(\tilde{K}_{t+1} + 2^{-j}\mathsf{B}) \leq 0.9 \cdot \mathsf{Vol}(K_t + 2^{-j}\mathsf{B})$$

which is the desired decrease in the $\Phi_t^j$ potential. This concludes the proof. $\qquad\square$

## 6 Local Contextual Recommendation

In this section, we consider the *local contextual recommendation* problem, in which we may choose a list of actions $L_t \subseteq \mathcal{X}_t$ and our feedback is some $x_t^{\text{loc}}$ such that $\langle x_t^{\text{loc}}, w^* \rangle \geq \max_{x \in L_t} \langle x, w^* \rangle$. In other words, the feedback may not be the optimal action but it must at least be as good as the local optimum in $L_t$. The goal is the same as before: minimize the total expected regret

$$\mathbb{E}[\text{Reg}] = \mathbb{E}\left[\sum_{t=1}^{T} \langle x_t^* - x_t, w^* \rangle \right] \text{ where } x_t^* \in \arg\max_{x \in \mathcal{X}_t} \langle x, w^* \rangle.$$

It should be noted that, in this model, it is impossible to achieve non-trivial regret if the list size $|L_t|$ is only one, since the feedback will always be the unique element, providing no information at all. Below we show that it is possible to achieve bounded regret algorithm even when $|L_t| = 2$, although the regret does depend on the total number of possible actions each round, i.e. $\max_t |\mathcal{X}_t|$. Furthermore, we show that, even when $|L_t|$ is allowed to be as large as $2^{\Omega(d)}$, the expected regret of any algorithm remains at least $2^{\Omega(d)}$.

### 6.1 Low-regret algorithms

We use $[a]_+$ as a shorthand for $\max\{a, 0\}$.

Our algorithm employs a reduction similar to that of Theorem 4. Specifically, we prove the following:

**Theorem 9.** *Suppose that $|\mathcal{X}_t| \leq A$ for all $t \in \mathbb{N}$, and let $H$ be any positive integer such that $2 \leq H \leq A$. Then, given a low-regret cutting-plane algorithm $\mathcal{A}$ with regret $\rho$, we can construct an $O(\rho \cdot A/(H-1))$-regret algorithm for local contextual recommendation where the list size $|L_t|$ in each step is at most $H$.*

Before we prove Theorem 9, notice that it can be combined with Theorem 5 and Theorem 6 respectively to yield the following algorithms for local contextual recommendation.

**Corollary 1.** *Suppose that $|\mathcal{X}_t| \leq A$ for all $t \in \mathbb{N}$, and let $H$ be any positive integer such that $2 \leq H \leq A$. Then, there is an $O\left(A/(H-1) \cdot \exp(d \log d)\right)$-regret algorithm for local contextual recommendation where the list size $|L_t|$ in each step is at most $H$.*

**Corollary 2.** *Suppose that $|\mathcal{X}_t| \leq A$ for all $t \in \mathbb{N}$, and let $H$ be any positive integer such that $2 \leq H \leq A$. Then, there is an $O(A/(H-1) \cdot d \log T)$-regret algorithm for local contextual recommendation where the list size $|L_t|$ in each step is at most $H$.*

Note that these algorithms work for list sizes as small as $H = 2$ but may also give a better regret bound if we allow larger lists.

We will now prove Theorem 9.

*Proof of Theorem 9.* Our algorithm is similar to that of Theorem 4, except that we also play $H - 1$ random actions from $\mathcal{X}_t$ in addition to the action determined by the answer of $\mathcal{A}$. More formally, each round $t$ of our algorithm works as follows:

- Ask $\mathcal{A}$ for its query $p_t$ to the separation oracle.
- Let $x_t = \text{BR}_t(p_t)$, and let $L_t' \subseteq \mathcal{X}_t$ be a random subset of $\mathcal{X}_t$ of size $\min\{H-1, |\mathcal{X}_t|\}$.
- Output the list $L_t = \{x_t\} \cup L_t'$.
- Let $x_t^{\text{loc}}$ be the feedback.
- If $x_t^{\text{loc}} \neq x_t$, do the following:
  - Return $v_t = (x_t^{\text{loc}} - x_t)/\|x_t^{\text{loc}} - x_t\|$ to $\mathcal{A}$.
  - Update the knowledge set $K_{t+1} = \{w \in K_t \mid \langle x_t^{\text{loc}} - x_t, w \rangle \geq 0\}$.

We will now show that the expected regret of the algorithm is at most $\rho \cdot A/(H-1)$. From the regret bound of $\mathcal{A}$, the following holds regardless of the randomness of our algorithm:

$$\rho \geq \sum_{t: x_t^{\text{loc}} \neq x_t} \left\langle \frac{x_t^{\text{loc}} - x_t}{\|x_t^{\text{loc}} - x_t\|}, w^* - p_t \right\rangle \geq \sum_{t: x_t^{\text{loc}} \neq x_t} 0.5 \left\langle x_t^{\text{loc}} - x_t, w^* - p_t \right\rangle$$

$$= 0.5 \left( \sum_t \left\langle x_t^{\text{loc}} - x_t, w^* - p_t \right\rangle \right).$$

From the requirement of $x_t^{\text{loc}}$, we may further bound $\langle x_t^{\text{loc}} - x_t, w^* - p_t \rangle$ by

$$\langle x_t^{\text{loc}} - x_t, w^* - p_t \rangle \geq \max_{x \in L_t} \langle x - x_t, w^* - p_t \rangle = \max_{x' \in L_t'} [\langle x' - x_t, w^* - p_t \rangle]_+.$$

Hence, from the above two inequalities, we arrive at

$$2\rho \geq \sum_t \max_{x' \in L_t'} [\langle x' - x_t, w^* - p_t \rangle]_+.$$

Next, observe that

$$\mathbb{E}\left[ \max_{x' \in L_t'} [\langle x' - x_t, w^* - p_t \rangle]_+ \right] \geq \Pr[x_t^* \in L_t'] \cdot \langle x^* - x_t, w^* - p_t \rangle$$

$$= \frac{|L_t'|}{|\mathcal{X}_t|} \cdot \langle x^* - x_t, w^* - p_t \rangle$$

$$\geq \frac{H-1}{A} \cdot \langle x^* - x_t, w^* - p_t \rangle.$$

Combining the above two inequalities, we get

$$2\rho \geq \frac{H-1}{A} \cdot \mathbb{E}\left[ \sum_t \langle x_t^* - x_t, w^* \rangle \right].$$

From this, we can conclude that the expected regret, which is equal to $\mathbb{E}[\sum_t \langle x_t^* - x_t, w^* \rangle]$, is at most $O(\rho \cdot A/(H-1))$ as desired. $\qquad\square$

## 6.2 Lower Bound

We will now prove our lower bound. The overall idea of the construction is simple: we provide an action set that contains a "reasonably good" (publicly known) action so that, unless the optimum is selected in the list, the adversary can return this reasonably good action, resulting in the algorithm not learning any new information at all.

**Theorem 10.** *Any algorithm for the local contextual recommendation problem that can output a list of size up to $2^{\Omega(d)}$ in each step incurs expected regret of at least $2^{\Omega(d)}$.*

*Proof.* Let $S$ be any maximal set of vectors in $B_d$ such that the first coordinate is zero and the inner product between any pair of them is at most $0.1$. By standard volume argument, we have $|S| \geq 2^{\Omega(d)}$. Furthermore, let $e_1$ be the first vector in the standard basis. Consider the adversary that picks $u \in S$ uniformly at random and let $w^* = 0.2e_1 + 0.8u$ and let $X_t = S \cup \{e_1\}$ for all $t \in \mathbb{N}$. The adversary feedback is as follows: if $u \notin L_t$, return $e_1$; otherwise, return $u$.

We will now argue that any algorithm occurs expected regret at least $2^{\Omega(d)}$, even when allows to output a list $L_t$ of size as large as $\lfloor \sqrt{|S|} \rfloor = 2^{\Omega(d)}$ in each step. From Yao's minimax principle, it suffices to consider only any deterministic algorithm $\mathcal{A}$. Let $L_t^0$ denote the list output by $\mathcal{A}$ at step $t$ if it had received feedback $e_1$ in all previous steps.

Observe also that in each step for which $u \notin L_t$, the loss of $\mathcal{A}$ is at least $0.6$. Furthermore, in the first $m = \lfloor 0.1\sqrt{|S|} \rfloor$ rounds, the probability that the algorithm selects $u$ in any list is at most $\frac{m\sqrt{|S|}}{|S|} \leq 0.1$. Hence we can bound the the expected total regret of $\mathcal{A}$ as:

$$\mathbb{E}[0.6 \cdot |\{t \mid u \notin L_t\}|] \geq 0.6m \Pr[u \notin \cup_{t=1}^m L_t] = 0.6m \Pr[u \notin \cup_{t=1}^m L_t^0] \geq 0.6m \cdot 0.9 \geq 2^{\Omega(d)}$$

which concludes our proof. $\qquad\square$