# OpenReview forum: "Contextual Recommendations and Low-Regret Cutting-Plane Algorithms"
_NeurIPS.cc/2021/Conference — NeurIPS 2021 Poster_

### Official Review · Reviewer_7k7P · 2021-07-16

**Rating:** 7
**Confidence:** 2

**Summary:**


This work studies a variant of contextual linear bandits, where for each episode the learner is presented with a set of possible actions, selects an action, incurs the loss, and observes the best action of the given action set instead of the loss function. The authors first relate the contextual recommendation problem to the low-regret cutting plane methods based on well-studied separation oracle methods, which have been developed to minimize the number of calls. The relationship is formally stated in Theorem 3.1.

Then, with the help of strong separation oracles, the authors design algorithms which achieve O(d log T) and O(exp(d log d)) regret based on existing cutting-plane algorithm, where d is the number of dimensions and T is the number of episodes.

Besides this contextual recommendation setting, the authors also consider the list contextual recommendation where the learner is allowed to recommend a bounded set of actions, and the local contextual recommendation where the feedback for each episode is a better action instead of the best action in the given action set.

**Limitations And Societal Impact:**

This work is pure theoretical and does not have any potential negative societal impact.

**Main Review:**

1. Contribution

The algorithm relies on specific geometric properties of the Steiner potential and the center-of-gravity such as that stated in Theorem 2.1. With the help of these properties and the algorithm of Liu et al. (2021), the authors proves that, by simply selecting the center-of-gravity of certain convex sets, one can achieve exp(O(d log d)) or O(d log T) regret for low-regret cutting plane problems.

To the best of my knowledge, the problem setups and the usage of geometric techniques are novel and realistic. On the other hand, the efficiently discretization of the curvature path is also new to me.

2. Weaknesses

The results are purely theoretical and I wonder if the algorithms can be efficiently implemented.

3. Writing Issues

The format of references seems incorrect.


**Time Spent Reviewing:**

25

---

> ### Author Response · Authors · 2021-08-11
> **Response**
>
>
> Thank you for your review. Our paper makes the first theoretical contribution for the problem of contextual recommendation. All the algorithms that we design / cite can be efficiently implemented in polynomial time. We will correct any mistakes in the reference format in future versions of this paper.

---

### Official Review · Reviewer_77iV · 2021-07-17

**Rating:** 7
**Confidence:** 1

**Summary:**

In the contextual recommendation (CR) problem, the agent receives in each round the best arm given the contextual, which is contrasted to the standard online learning procedure where a reward is usually observed. To tackle the problem, this paper explores the connections between CR and the cutting-plane algorithms, which aims to locate a hidden point by iterative hyperplanes. The main theorem in the paper states that a $O(\tho)$-regret CR algorithm can be constructed from a $O(\rho)$-regret cutting-edge algorithm. The paper further proposes two variants of algorithms that achieve low regrets. Furthermore, the paper tackles other variants of the CR problem and presents algorithms with guaranteed regrets. In general, this paper is a theoretical paper. I think the theorems and the algorithms can be important to the field.

**Main Review:**

Although the paper aims mainly on the theoretical side, adding empirical verifications can be more convincing and illustrative.

Question: when the separation oracle is weak, i.e., the direction is chosen before the agent makes the choices, the resulting regret is worse (Theorem 5.2) than Theorem 3.1. Why a weaker oracle leads to a more difficult problem?

**Time Spent Reviewing:**

8

---

> ### Author Response · Authors · 2021-08-11
> **Response**
>
>
> Thank you for your response. For the case of a separation oracle that is weak, we obtain results that are polynomial in the dimension. For the case of the strong separation oracle, our results have an exponential dependence on d or they have a dependence on poly(T) (which could be arbitrarily large as a function of d).

---

### Official Review · Reviewer_oGv3 · 2021-07-19

**Rating:** 7
**Confidence:** 3

**Summary:**

This paper introduces the problem of low-regret cutting plane algorithms and studies its applications to contextual recommendations, which is also introduced in this paper. The low-regret cutting plane problem is similar to the standard cutting plane setting, but instead of bounding the number of iterations the goal is to bound the total regret. Specifically, we have a convex body $K_1$ and a hidden point $w^* \in K_1$. In round $t\geq 1$, we pick a point $p_t\in K_t$, the adversary picks a direction $v_t$ such that $\langle w^* - p_t, v_t\rangle \geq 0$, and the next convex body $K_{t+1}$ is defined as the intersection of $K_t$ with the halfspace $\{x : \langle x - p_t, v_t\rangle \geq 0\}$. The goal is to minimize the total regret $\sum\limits_{t=1}^{T} \langle w^* - p_t, v_t\rangle$. In the convextual recommendation problem, there is again a hidden point $w^*$ (corresponding e.g. to some unknown user preferences). In each step $t$, the goal is to pick some $x_t$ from a set of possible actions $\mathcal{X}_t$ (e.g. set of all paths in route planning). Then, we pay $\langle x_t^* - x_t, w^* \rangle$, where $x_t^*$ is the optimal action and subsequently get to learn $x_t^*$. This is different from other known settings where the feedback is instead the value of the regret, i.e. $\langle x_t^* - x_t, w^* \rangle$.

The main results are:
* An $exp(O(d\log d))$-regret cutting plane algorithm
* An $O(d\log T)$-regret cutting plane algorithm
* An $O(\mathrm{poly}(d))$-regret cutting plane algorithm if the adversarial direction $v_t$ does not depend on the point $p_t$ chosen.

The first two immediately transfer to the contextual recommendation problem, and the third one can be translated to an algorithm for a weaker variant of contextual recommendation where there are $\mathrm{poly}(d)$ recommendations instead of $1$.

The algorithms are based on convex geometry, and the techniques include the Steiner potential and a new characterization of the centroid of $K+rB$ (i.e. all points within distance $r$ from $K$) as a function of $r$.

**Limitations And Societal Impact:**

* No numerical experiments

**Main Review:**

I like the fact that the authors define the low regret cutting plane problem in an abstract way, as this might help attract some attention to the problem and motivate further connections to other problems. The idea of bounding the length of the centroid curve is nice and new, to the best of my knowledge, and it takes another step into showing how useful Steiner polynomials are for these types of problems. The first two low-regret cutting plane algorithms are more standard, and their techniques follow more naturally from previous work. In particular, these (especially the $O(d\log T)$ bound) seem quite similar to (and maybe even recoverable by) [supp 20]. It would be useful if the authors could clarify if this is not the case. Even so, the authors' approach of dealing with the parametric convex body $K+rB$ seems slightly cleaner. More generally, it would be good to clarify how contextual recommendations and contextual search are different, in terms of the techniques used in them. I have the following comments:

* I would be interested in getting more thorough motivation behind the contextual recommendation problem. In particular, the fact that it is assumed that the user will pick the _optimal_ action seems somewhat less natural to me than the setting where the user can pick any action that is at least as good as the suggestions. Unfortunately, the bounds for the latter depend linearly on the ratio $|\mathcal{X}_t| / (\text{number of suggestions} - 1)$, which seems impractical ($\mathcal{X}_t$ could be very large, e.g. the total number of paths in a graph).
* Related to the previous point, it would be useful to have some experiments of the proposed algorithms on real data, as the problem of contextual recommendations is very practically motivated. In particular, this might show that the bounds for the latter (more realistic) setting from the previous bullet are too pessimistic, which would be interesting.
* There is something off in the paragraph starting in line 314. If we incur loss $w\approx d r$, and this happens $O(d\log (2/r))$ times, then the total loss should at
best be $O(d^2)$ and not $O(d\log d)$.

I think the main significance of this work is formalizing the low-regret cutting plane algorithms, and introducing the centroid curve in the context of cutting plane algorithms.

Some typos (from the supplementary):
* Line 82: What is K?
* Line 168: Minkowsi
* Lines 231-232: doesn’t make sense
* Line 419: The quantity right after this line is negative
* It seems to be implied that K_1 = B, but I couldn't find where it was stated
* Line 463: I think it should be (2T)^d

**Time Spent Reviewing:**

2

---

> ### Author Response · Authors · 2021-08-11
> **Response**
>
> Thank you for your response.  Although we motivated our problem in the setting of finding the best route in a geographical query service, we believe that the general problem can be well motivated in a variety of settings. Consider any setting where a student (say a junior doctor, new lawyer etc) has a number of different actions to take when presented in a case. They may suggest the best course which is then corrected by a teacher (say a senior doctor or experienced lawyer). One may not be able to know the exact reward but one is told the best action to take. This type of learning is precisely captured by the contextual recommendation problem.
>
> We note that previous results on contextual search (esp [20]) rely explicitly on knowing the direction vector v_t before picking the point x_t. When this direction can be adversarially chosen, then the algorithms do not give any bounded regret guarantees. This is one of the main difficulties in the problem of contextual recommendation.
>
> The final point can be clarified as r is on the scale of O(1/d) and as a result the resulting loss is proportional to O(d^2r log d)= O(d log d). We will clarify this and the other corrections you mentioned in future versions of the paper.

---

### Official Review · Reviewer_1hLE · 2021-07-21

**Rating:** 7
**Confidence:** 4

**Summary:**

Authors address a variant of a contextual linear bandit problem where the objective is to estimate the unknown d-dimension vector w* by only observing the identity of the best arm (from the context set X_t) after each round. The paper design algorithms for this problem that achieve regret O(d log T) and exp(O(d log d)) using novel cutting-plane algorithms. They also studied a variant that allowed to provide a list of several recommendations and give an algorithm with O(d2 log d) regret, list size being O(poly(d)).

**Limitations And Societal Impact:**

Satisfactory.

**Main Review:**

The paper suggests a novel cutting-plane algorithm-based method to address the problem of contextual search problem. The problem formulation is novel and relevant and their proposed algorithms provide near-optimal bounds for the objective.
Is not the exp(d) regret bound trivially follows as there can only be O(2^d) "distinct X_ts" possible? What happens if X_t's are generated stochastically from some underlying set distribution, instead of adversarially? Instead of the identity of the best arm, if a ranking over first m best arms (m < A) are revealed, how do you expect the regret bound to be scaled. Would the results if the potential class of X_ts are infinite / continuous (in other words, what is the notion of gap that shows up in the regret bounds)?

**Time Spent Reviewing:**

~3-4

---

> ### Author Response · Authors · 2021-08-11
> **Response**
>
> Thank you for your thoughtful response. We want to clarify that our algorithms do not need the action set to be finite. As a result, it is not the case that a O(2^d) regret follows as there are many more actions available. This also makes it difficult to define what the m best arms are.
>
> Although our algorithm provides the same guarantees in the stochastic setting, it is an interesting question if the bounds can be improved under a distributional assumption.

---

> > ### Comment · Reviewer_1hLE · 2021-08-27
> > **Post rebuttal**
> >
> > Author responses are satisfactory, I keep my score unchanged and vote for accepting the submission.

---

### Decision · Program_Chairs · 2021-09-28

**Decision:**

Accept (Poster)

**Comment:**

This paper introduces the problem of contextual recommendation, a variant of the linear contextual bandit algorithm in which a learner selects actions based on context in order to maximize reward, but rather observing the reward, observe the identity of the optimal action instead. The reviewers found that the results are novel and technically interesting, and in particular thought that formalizing the problem of regret for cutting plane algorithms and connecting this to contextual recommendation was valuable, and is likely to find further use. Overall, this is a solid contribution (albeit, slightly niche for the NeurIPS community). The authors are encouraged to incorporate the reviewers' suggestions and spend more time motivating the problem, as well as discussing practical aspects of their algorithms (e.g., implementation).

**Consistency Experiment:**

NeurIPS has a long history of experimentation. In 2014, NeurIPS ran an experiment in which 10% of submissions were reviewed by two independent committees to quantify the randomness in the review process. This year, we repeated a variant of this experiment to see how the quality of the review process has changed over time.  This paper was part of the experiment and was therefore assigned to two committees (consisting of reviewers, an Area Chair, and a Senior Area Chair) that reached independent decisions.  If both committees made the same recommendation, this recommendation was followed. If a single committee recommended acceptance, the paper was accepted (with the exception of a few cases in which the other committee identified what we considered a fatal flaw, e.g., an error in a key result).

This copy’s committee reached the following decision: **Accept (Poster)**

The other committee assigned to the paper recommended **Reject**.  You can find the other set of reviews, along with any follow up discussion with the authors here:
https://openreview.net/forum?id=hVnM-ni5o5nVQ